# HFT: Half Fine-Tuning for Large Language Models

## Abstract

Large language models (LLMs) with one or more fine-tuning phases have become necessary to unlock various capabilities, enabling LLMs to follow natural language instructions and align with human preferences. However, it carries the risk of catastrophic forgetting during sequential training, the parametric knowledge or the ability learned in previous stages may be overwhelmed by incoming training data. This paper finds that LLMs can restore some of the original knowledge by regularly resetting partial parameters. Inspired by this, we introduce Half Fine-Tuning (HFT) for LLMs, as a substitute for full fine-tuning (FFT), to mitigate the forgetting issues, where half of the parameters are selected to learn new tasks. In contrast, the other half are frozen to retain previous knowledge. We provide a feasibility analysis from the perspective of optimization and interpret the parameter selection operation as a regularization term. Without changing the model architecture, HFT could be seamlessly integrated into existing fine-tuning frameworks. Extensive experiments and analysis on supervised fine-tuning, direct preference optimization, and continual learning consistently demonstrate the effectiveness, robustness, and efficiency of HFT. Compared with FFT, HFT not only significantly alleviates the forgetting problem, but also achieves the best performance in a series of downstream benchmarks, with an approximately 30% reduction in training time.

## 1 Introduction

Large language models (LLMs) bring immense revolutions to various natural language processing applications with powerful language understanding and generation capabilities. Unsupervised large-scale pre-training for learning basic world knowledge (hereinafter referred to as basic knowledge), followed by one or more fine-tuning phases with supervised data or human feedback, is becoming a new training paradigm in the era of LLMs (Ouyang et al., 2022; Achiam et al., 2023; Touvron et al., 2023). As the fine-tuning phase proceeds, the enormous potential of LLMs is gradually unleashed to handle various downstream tasks, while the parametric knowledge previously learned and stored in the pre-trained model might face a considerable risk of *catastrophic forgetting* (Lin et al., 2024; Neeman et al., 2023; Dong et al., 2024). To maintain intrinsic basic knowledge, the most straightforward idea is to keep the pre-trained parameters unchanged and include extra modules to learn task-specific abilities (Dou et al., 2023; Wu et al., 2024a). However, such architectural modifications pose significant obstacles to model deployment and continual fine-tuning.

Without changing model architecture, vanilla full fine-tuning (FFT) methods update all parameters to improve the performance of downstream tasks (Zhang et al., 2023c), in which the element-wise parameter difference between fine-tuned and pre-trained models (i.e., task vector) represents the knowledge shift during fine-tuning (Ilharco et al., 2023). Herein, a desirable task vector is expected to keep basic knowledge of the pre-trained model and learn new specialized knowledge. Interestingly, recent work shows that partial dropping or trimming of the task vector has only milder impacts on target task (Yadav et al., 2023; Yu et al., 2023). In other words, partial new parameters are sufficient for the learning of new abilities, so the upcoming question is, *is it possible that a portion of old parameters could maintain the capabilities of the pre-trained model?*

To answer this question, we start with Llama 2-7b and Llama 2-Chat-7b, and attempt to reset partial parameters of the chat-model to the pre-trained model, then prob the general abilities and basic knowledge of these models (see Figure 1). As a representative general-purpose fine-tuning practice,

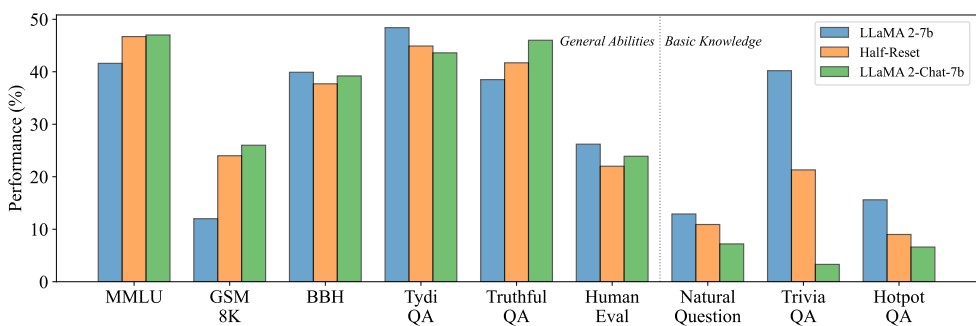

Figure 1: Performance of LLAMA 2-7B, LLAMA 2-CHAT-7B, and the *Half-Reset* model on six general abilities and three basic knowledge benchmarks. It is interesting that simply resetting half of the parameters of the chat-model to the pre-trained model could roughly restore a significant amount of forgotten basic knowledge while maintaining high-level general abilities performance.

there is some improvement in the general abilities of LLAMA 2-CHAT-7B, while the basic knowledge falls off a cliff. It is consistent with previous observations, indicating the destruction of parametric knowledge stored in LLAMA 2-7B (Dou et al., 2023). To balance the emerging general abilities and the inherent basic knowledge, we intuitively select and reset *half of the parameters*[1] of LLAMA 2-CHAT-7B and are pleasantly surprised to find that the *Half-Reset* model greatly resumes the basic knowledge in LLAMA 2-7B while remaining the excellent general abilities of LLAMA 2-CHAT-7B (More details in Section 2).

Inspired by these above observations, we propose Half Fine-Tuning (HFT), a simple yet effective approach for the training of LLMs and further extrapolate it to the continual fine-tuning scenarios. Specifically, in each round of fine-tuning, we randomly select and freeze half of the parameters, and only update the other half, which allows the model to retain the ability of the startup point while learning downstream tasks. Note that HFT does not change the model architecture or traditional fine-tuning paradigm, thus theoretically it can be applied to any setting where the standard full fine-tuning is previously applicable, including but not limited to supervised fine-tuning (SFT), direct preference optimization (DPO), continual learning (CL), etc.

To evaluate the effectiveness of HFT in instruction fine-tuning settings, we conduct extensive experiments with TÜLU V2 (Ivison et al., 2023) for SFT and UltraFeedback (Cui et al., 2023) for DPO. Simultaneously, we also extend experiments on TRACE (Wang et al., 2023a) for CL (i.e. multi-round fine-tuning) to validate the proposed method in a more extreme scenario. Experimental results demonstrate that HFT not only exhibits excellent talent in alleviating catastrophic forgetting but also achieves comparable or even better performance in learning new abilities compared to FFT. Further analysis reveals that regardless of which half (or even only about half) of the parameters are selected, HFT is capable of attaining tolerable performance gains and impressive efficiency improvements, which brings considerable competition to the routine fine-tuning paradigm.

In summary, the main contributions of this paper are as follows:

- We reveal that by resetting half of the fine-tuned parameters to the startup state, it is possible to preliminary restore the primaeval ability while maintaining new learning ability, which poses new opportunities to alleviate catastrophic forgetting and obtain an all-around LLM.

- We propose Half Fine-Tuning (HFT), which entails freezing half of the parameters while training the other half. It allows LLMs to acquire new abilities while retaining and utilizing previously learned knowledge in various training settings.

- Extensive experiments and analysis demonstrate the effectiveness and efficiency of HFT. Without any alterations to the model architecture, HFT, as a plug-and-play solution with only a few lines of code, exhibits the potential to supersede FFT in the era of LLMs.

---

[1]Here, we keep the `embedding` and `lm_head` layers unchanged as LLAMA 2-CHAT-7B, and select 50% of the parameters in `transformer` layers. The parameter ratios in this paper all follow this statistical calibre.

## 2 PILOT EXPERIMENTS

Considering that the partial task vector is capable of maintaining new abilities (Yadav et al., 2023; Yu et al., 2023), we attempt to roll back the primaeval abilities of pre-trained models by resetting the remaining part of the task vector, thereby alleviating the catastrophic forgetting problem caused by fine-tuning. In this section, We employ the representative well-aligned LLM, LLAMA 2-CHAT-7B, and the corresponding pre-trained backbone, LLAMA 2-7B, as models for analysis.

**Setup.** To balance the original abilities and the enhanced capabilities gained through instruction tuning, we simply choose to reset 50% of the parameters in LLAMA 2-CHAT-7B to LLAMA 2-7B, so that half of the parameters are hoped to align with the new tasks, while the other half is intended to restore the old capabilities. In the implementation, we randomly select half of each transformer layer according to the category of the parameter matrix. Specifically, we choose two from four self-attention matrices (i.e., $\mathbf{W}_Q, \mathbf{W}_K, \mathbf{W}_V, \mathbf{W}_O$), and for the odd parameter number in LLAMA's feed-forward layers (i.e., $\mathbf{W}_{up}, \mathbf{W}_{down}, \mathbf{W}_{gate}$), we randomly select half of the transformer layers to choose two matrices and the other half to choose one. Such a fine-grained selection strategy ensures that the *Half-Reset* operation rolls back exactly 50% of the parameters.

To assess the performance of the pre-trained, chat, and half-reset models on both new and old capabilities, we follow Ivison et al. (2023) and Dou et al. (2023) to introduce two categories of evaluation benchmarks: (1) **General Abilities**, including MMLU, GSM8K, BBH, TyDiQA, TruthfulQA, and HumanEval, which measure the LLMs' newly enhanced abilities to perform specific downstream tasks like examination, reasoning, and coding. (2) **Basic Knowledge**, including NaturalQuestion, TriviaQA, and HotpotQA, which reflect the parametric world knowledge in the pre-trained model and could be used to evaluate retention of the primaeval capabilities. For more details about the datasets and evaluation metrics, please refer to Appendix A.2.1 and A.2.2

**Results.** From Figure 1, it is intuitive to observe significant improvement of LLAMA 2-CHAT-7B on several general ability benchmarks, as well as the comprehensive decline on the basic knowledge benchmarks. When selectively restoring half parameters to the pre-trained LLAMA 2-7B model, although there is a slight performance loss in the overall performance of general abilities, we witness the remarkable recovery of basic knowledge. In Appendix A.3.1, we attempt other possible half-reset solutions and provide more numerical results, all of which exhibit similar phenomena.

In conclusion, the pilot experiments demonstrate that (1) full parameter fine-tuning with large-scale instruction data disrupts the basic knowledge stored within pre-trained LLMs. (2) Through a simple half-reset operation, it is possible to restore the forgotten knowledge partially. *Take another step forward, these findings open a new door for model merging, inspiring us to preserve some mastered abilities of the startup point by freezing partial parameters during fine-tuning.*

## 3 METHODOLOGY

Without loss of generality, we consider a sequential (continual) learning setting with multiple tasks $\mathcal{T}$, in which each task corresponds to a set of input-output pairs $\mathcal{D}^t = \{x_n^t, y_n^t\}_{n=1}^{\mathcal{N}^t}$. In the training process, a single model aligns all the tasks sequentially, with only access to the specific dataset $\mathcal{D}^t$ at $t$-th round. Formerly, given an LLM parameterized by $\theta$, the entire process aims to optimize the following objective, which encompasses all the tasks,

$$\mathcal{J}(\theta) = \max_{\theta} \sum_{t \in \{1, |\mathcal{T}|\}} \sum_{(x_n^t, y_n^t) \in \mathcal{D}^t} \log \mathbf{P}_{\theta^t}\left(y_n^t | x_n^t\right), \tag{1}$$

where $\log \mathbf{P}(\cdot)$ represents the probability distribution of the model's output. When there is only one task, the learning process degenerates into the standard supervised fine-tuning (SFT) form.

**Half Fine-Tuning.** Next, we accordingly propose Half Fine-Tuning (HFT) to learn the upcoming new task while maintaining and utilizing old abilities. Figure 2 illustrates the overall workflow of HFT, regarding the intermediate repetitive transformer layers, we divide each layer into three blocks: self-attention, feed-forward, and layernorm, so as half of each block is selected for updating in this round, while the remaining half is frozen. Note that the frozen and updated parameters vary among each training round. In this way, HFT is more conducive to maintaining relative knowledge parity across different rounds during the sequential alignment process, thus exhibiting significant scalability

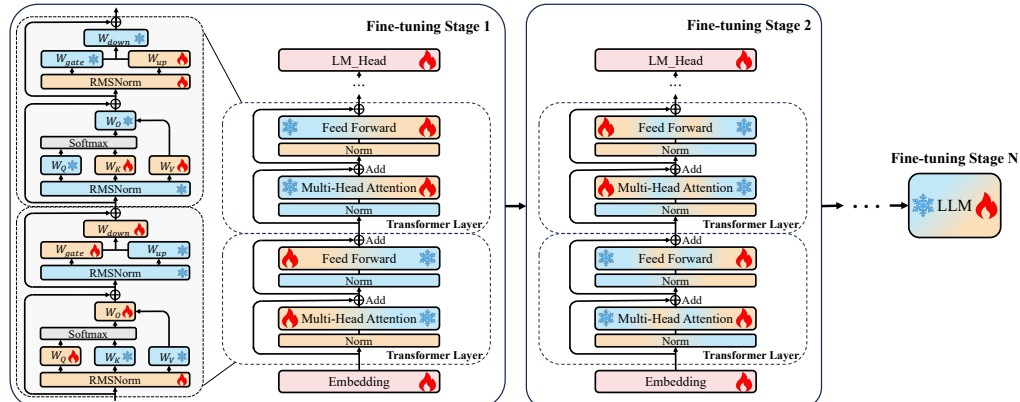

Figure 2: The schematic procedure of HFT with LLAMA 2's architecture. In each stage, we selectively freeze half of the parameters at the category-level and update the other half. Best viewed in colour.

in successive training. From the formula perspective, we define the parameters that remain unchanged during the $t$-th round as $\psi^t$, and correspondingly, the parameters that align to the upcoming tasks as $\vartheta^t$ (i.e., $\theta^t = \{\vartheta^t, \psi^t\}$). The training objective in Equation 1 thus changes to

$$\mathcal{J}(\theta) = \max_\theta \sum_{t \in \{1, |\mathcal{T}|\}} \sum_{(x_n^t, y_n^t) \in \mathcal{D}^t} \log \mathbf{P}_{\{\vartheta^t, \psi^t\}}\left(y_n^t | x_n^t\right),$$

$$s.t. \quad \vartheta^t \leftarrow \vartheta^{t-1} - \eta \nabla_\vartheta \mathcal{L}\left(\theta^{t-1}\right), \quad \psi^t \leftarrow \psi^{t-1}, \tag{2}$$

where $\eta$ and $\mathcal{L}(\cdot)$ represent the learning rate and loss function, $\nabla_\vartheta$ indicates that we only consider the gradients of selected parameters in fine-tuning.

**Why Half Fine-Tuning Works.** Excluding heuristic motivations, we are also interested in the theoretical principles behind HFT. Theoretically, HFT could be regarded as exerting a parameter-level mask to vanilla FFT. In this part, we borrow the thread in Fu et al. (2022) to interpret why HFT works from the perspective of optimization. Given a pre-trained model $\mathcal{M}^0$ with parameters $\theta^0$, the fine-tuned model $\mathcal{M}$ with parameters $\theta$ has the same structure as $\mathcal{M}^0$ such that $\|\theta - \theta^0\|_0 \leq p \dim(\theta)$, where $p = 0.5$ in HFT. Next, we denote $M \in \{0, 1\}^{m \times m}$ as a mask diagonal matrix on the parameter, in which the diagonal is equal to 1 if the parameter is selected, thus the fine-tuning procedure can be formulated as $\theta = \theta^0 + M\Delta\theta$, where $\Delta\theta$ is the task vector. In that case, HFT solves an optimization problem with constraints $\min_{\Delta\theta, M} \mathcal{L}(\theta^0 + M\Delta\theta)$ such that $\|M\|_0 = \lfloor mp \rfloor$; $M_{ij} = 0, \forall i \neq j$; $M_{ii} \in \{0, 1\}$. where $\mathcal{L}$ is the loss function, $\lfloor \cdot \rfloor$ is the floor function, $m$ is the parameter numbers. By integrating previous conditions, the optimization procedure of HFT can be reformulated as

$$\mathcal{O} = \min_\theta \mathcal{L}(\theta) \quad s.t. \quad \|(I - M)(\theta - \theta^0)\|^2 = 0, \tag{3}$$

With Lagrangian duality, solving the constrained optimization problem is equivalent to solving the following unconstrained optimization problem

$$\mathcal{O}_L = \min_\theta \max_\lambda \mathcal{L}(\theta) + \lambda \|(I - M)(\theta - \theta^0)\|^2, \tag{4}$$

where $\lambda$ is the Lagrange multiplier. Based on the Minimax inequality, it is intuitive to derive that $\min_\theta \max_\lambda \mathcal{L}(\theta) + \lambda \|(I-M)(\theta-\theta^0)\|^2 \geq \max_\lambda \min_\theta \mathcal{L}(\theta) + \lambda \|(I-M)(\theta-\theta^0)\|^2 \geq \min_\theta \mathcal{L}(\theta) + \|(I-M)(\theta-\theta^0)\|^2$. In conclusion, the optimization process of HFT is equivalent to optimizing the upper bound of the FFT loss function $\mathcal{L}(\theta)$ with a regularization term $\|(I-M)(\theta-\theta^0)\|^2$. From the optimization perspective, such regularization (with an appropriate sparsity $M$) contributes to the stability of the sparse fine-tuned model (Radiya-Dixit & Wang, 2020; Fu et al., 2022), meaning that HFT has the opportunity to achieve results comparable to or even better than FFT, theoretically.

## 4 EXPERIMENTS

In this section, we primarily report the experimental results of full fine-tuning (FFT) and the proposed half fine-tuning (HFT) on supervised fine-tuning (with TÜLU V2 (Ivison et al., 2023) as training set),

Table 1: Results on general ability benchmarks of various models with instruction tuning (SFT, DPO), in which the default setting is FFT, R and H refer to the proposed Half-Reset and Half Fine-Tuning methods, respectively. Bold text denotes the best result in each group. More baselines in Table 8.

| | MMLU (factuality) | GSM8K (reasoning) | BBH (reasoning) | TyDiQA (multilingual) | TruthfulQA (truthful) | HumanEval (coding) | Overall |
|---|---|---|---|---|---|---|---|
| | EM (0-shot) | EM (8-shot, CoT) | EM (3-shot, CoT) | F1 (1-shot, GP) | MC2 (0-shot) | Pass@10 (0-shot) | |
| *Pre-trained models* | | | | | | | |
| LLAMA 2-7B | 41.6 | 12.0 | 39.9 | 48.4 | 38.5 | 26.2 | 34.4 |
| LLAMA 2-13B | 52.2 | 34.5 | 50.7 | 50.3 | 49.8 | 32.7 | 45.0 |
| *Supervised Fine-Tuning (SFT) on* TÜLU *V2* | | | | | | | |
| LLAMA 2-7B-SFT | 48.5 | 25.0 | 42.2 | 51.2 | 41.7 | **36.9** | 41.0 |
| LLAMA 2-7B-SFT (R) | 48.4 | 23.0 | 43.4 | **52.4** | 42.5 | 32.5 | 40.4 |
| LLAMA 2-7B-SFT (H) | **50.8** | **30.5** | **43.6** | 52.3 | **45.4** | 34.6 | **42.9** (+1.9) |
| LLAMA 2-13B-SFT | 50.6 | 45.0 | 47.8 | 55.0 | 42.6 | 42.4 | 47.2 |
| LLAMA 2-13B-SFT (R) | 52.7 | 46.0 | 52.8 | 55.5 | **46.8** | 41.4 | 49.2 |
| LLAMA 2-13B-SFT (H) | **54.5** | **46.5** | **53.7** | **56.7** | 45.7 | **43.5** | **50.1** (+2.9) |
| *Direct Preference Optimization (DPO) on UltraFeedback* | | | | | | | |
| LLAMA 2-7B-DPO | 48.9 | 28.0 | 42.9 | 50.2 | **45.7** | 35.6 | **41.9** |
| LLAMA 2-7B-DPO (R) | **49.0** | **28.5** | **43.1** | 50.3 | 43.3 | 34.8 | 41.5 |
| LLAMA 2-7B-DPO (H) | 48.8 | 25.5 | 42.8 | **51.1** | 45.5 | **36.7** | 41.7 (-0.2) |
| LLAMA 2-13B-DPO | **52.0** | 44.0 | 47.1 | 51.5 | 45.5 | **44.3** | 47.4 |
| LLAMA 2-13B-DPO (R) | 51.5 | 46.5 | 48.2 | **53.7** | 43.7 | 42.7 | 47.7 |
| LLAMA 2-13B-DPO (H) | 51.8 | **48.5** | **49.9** | 52.9 | 45.3 | 41.0 | **48.2** (+0.8) |

human preference alignment (with UltraFeedback (Cui et al., 2023)), and continual learning (with TRACE (Wang et al., 2023a)) scenarios, in which direct preference optimization (DPO) (Rafailov et al., 2023) is used to learn human preferences. Following Ivison et al. (2023) and Wang et al. (2023a), we employ LLAMA 2 and LLAMA 2-CHAT as the backbone model, respectively. Apendix A.2 shows more information about implementations and Appendix A.3 proposes more additional experiments consist of the comparison with more baselines, the impact of learning rates and random seeds, the exploration of DPO on HFT-based models, efficiency analysis and many other detailed results.

### 4.1 EXPERIMENTS ON INSTRUCTION TUNING

**Setup.** We employ the general abilities and basic knowledge benchmarks mentioned in Section 2 to evaluate various models under the instruction tuning settings. In Appendix A.3.2, we introduce a series of sparse fine-tuning and model merging methods as additional baselines. To assess the conversation ability, we also compare these models on AlpacaEval 2.0 (see Appendix A.3.8).

**Results on Improving General Abilities.** Results in Table 1 demonstrate the effectiveness of our proposed HFT method, which simultaneously improves different specialized abilities by selectively fine-tuning half of the parameters. Specifically, compared to FFT under the SFT setting, HFT leads to an overall performance improvement of 1.9% on LLAMA 2-7B and 2.9% when scaling to LLAMA 2-13B. Furthermore, as we continue to perform DPO on SFT models, we observe that updating the policy model with HFT does not hinder the model from learning human preferences. *In sum, the HFT method has strong robustness to adapt to different fine-tuning algorithms.* Besides, we also attempt to review the *Half-Reset* method in Section 2, but the benefits of this approach are not robust, and we attribute it to the randomness of parameter operations. In comparison, HFT achieves a more stable performance improvement through the learning process, while avoiding the complexity of the two-stage process of fully updating followed by partially resetting.

**Results on Preserving Basic Knowledge.** When it comes to basic knowledge, as depicted in Table 2, both SFT and DPO exhibit a significant decline across all three benchmarks. *Notably, HFT demonstrates excellent talent in preserving basic knowledge, consistently outperforming fully updating parameters during SFT and DPO.* For example, during the SFT stage, HFT achieves improvements of 3.4% and 2.9% with LLAMA 2-7B and LLAMA 2-13B compared to FFT, respectively. It is worth mentioning that Half-Reset also shows a stable performance in alleviating knowledge forgetting, which once again confirms the motivation to keep partial initial parameters unchanged.

Table 2: Results on basic knowledge benchmarks of various models with instruction tuning.

| | NaturalQuestion (EM, 0-shot) | TriviaQA (EM, 0-shot) | HotpotQA (EM, 0-shot) | Overall |
|---|---|---|---|---|
| *Pre-trained models* | | | | |
| LLAMA 2-7B | 12.9 | 40.2 | 15.6 | 22.9 |
| LLAMA 2-13B | 9.6 | 24.0 | 13.4 | 15.7 |
| *Supervised Fine-Tuning (SFT) on TÜLU V2* | | | | |
| LLAMA 2-7B-SFT | 3.2 | 26.4 | 14.5 | 14.7 |
| LLAMA 2-7B-SFT (R) | **7.3** | 26.4 | 14.4 | 16.0 |
| LLAMA 2-7B-SFT (H) | 6.2 | **32.8** | **15.4** | **18.1** (+3.4) |
| LLAMA 2-13B-SFT | 0.7 | 9.2 | 4.9 | 4.9 |
| LLAMA 2-13B-SFT (R) | 1.8 | 13.5 | 5.3 | 6.9 |
| LLAMA 2-13B-SFT (H) | 2.7 | 12.4 | 8.2 | **7.8** (+2.9) |
| *Direct Preference Optimization (DPO) on UltraFeedback* | | | | |
| LLAMA 2-7B-DPO | 1.4 | 20.8 | 10.0 | 10.7 |
| LLAMA 2-7B-DPO (R) | **2.0** | **23.6** | 12.1 | **12.6** |
| LLAMA 2-7B-DPO (H) | 1.9 | 22.9 | **12.8** | 12.5 (+1.8) |
| LLAMA 2-13B-DPO | 0.1 | 4.4 | 2.4 | 2.3 |
| LLAMA 2-13B-DPO (R) | **0.3** | **6.5** | **3.8** | **3.5** |
| LLAMA 2-13B-DPO (H) | 0.2 | 5.5 | 3.0 | 2.9 (+0.6) |

***Remark.*** *HFT not only effectively preserves a certain degree of basic knowledge of the pre-trained model, but also utilizes this knowledge to achieve better learning of new abilities.*

## 4.2 EXPERIMENTS ON CONTINUAL LEARNING

**Setup.** We evaluate the performance in the continual learning setting (with TRACE (Wang et al., 2023a)), using four representative approaches and attempt to replace FFT with HFT. (1) **SeqFT**: It is a standard for sequentially learning all parameters of downstream tasks. (2) **GEM** (Lopez-Paz & Ranzato, 2017): It leverages episode memories to avoid forgetting, but it consumes extra computation time like other regularization-based methods. (3) **Replay**: It is a common strategy, here we integrate alignment data from LIMA (Zhou et al., 2023) into the replay memory and replaying 10% of historical data. (4) **LoraSeqFT** (Hu et al., 2022): It sequentially updates the low-rank matrices while keeping the backbone fixed. Note that the LoRA-based method modifies the model architecture and is not suitable for combination with HFT. Following (Wang et al., 2023a), we start with LLAMA 2-CHAT-7B/13B, adopt **Overall Performance (OP)** and **Backward Transfer (BWT)** as the evaluation metrics (Appendix A.2.2 details the calculation process). Besides, we also report the general abilities and basic knowledge of various models after the final round of learning (see Appendix A.3.4).

**Results.** Table 3 shows that the *three FFT approaches could all benefit from equipping HFT*. Specifically, HFT brings performance improvements of 5.7% and 2.0% on the OP metric in the SeqFT and GEM settings, respectively. It also boosts the performance with 4.6%, 0.7%, and 2.0% on the BWT metric based on the LLAMA 2-CHAT-7B. When scaling the model to 13b, HFT could also achieve superior performances. Further, fine-tuning with full parameters often suffers from severe catastrophic forgetting in the 5-th round (see Appendix A.3.11), while HFT does not experience such a problem in any of the rounds, making the learning process more stable. Besides, LoraSeqFT exhibits notably suboptimal performance in this setting. We assume that the knowledge capacity of the LoRA parameter is quite limited, thus resulting in considerable forgetting during the process of sequential training. On the contrary, HFT is based on a full set of parameters and selects half of the parameters to be fine-tuned in each round, which has a stronger knowledge tolerance.

Table 3: OP and BWT on TRACE with different strategies, OP measures the learning of new tasks and BWT measures the forgetting of old tasks.

| | FFT | | HFT | |
|---|---|---|---|---|
| | OP | BWT | OP | BWT |
| **LLAMA 2-CHAT-7B** | | | | |
| LoraSeqFT | 6.4 | -45.2% | - | - |
| SeqFT | 45.7 | -10.2% | 51.3 (+5.6) | -5.6% (+4.6) |
| GEM | 48.2 | -7.9% | 50.2 (+2.0) | -5.9% (+2.0) |
| Replay | 54.3 | 1.4% | 54.1 (-0.2) | +2.1% (+0.7) |
| **LLAMA 2-CHAT-13B** | | | | |
| LoraSeqFT | 26.5 | -30.0% | - | - |
| SeqFT | 49.0 | -9.4% | 52.0 (+3.0) | -8.5% (+0.9) |
| GEM | 50.4 | -8.9% | 53.6 (+3.2) | -6.1% (+2.8) |
| Replay | 54.7 | -0.6% | 57.4 (+2.7) | +1.6% (+2.2) |

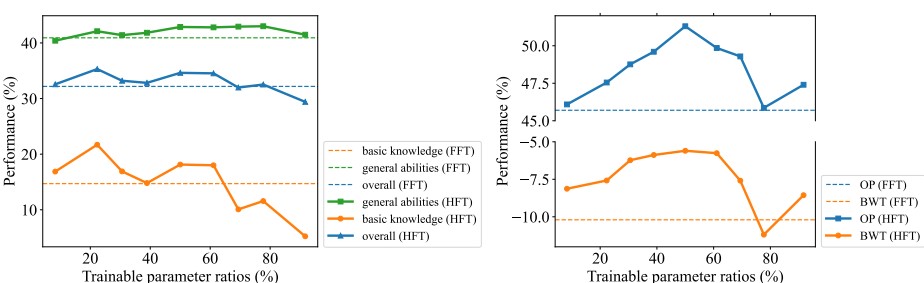

(a) Performance of models trained on TÜLU V2.  (b) Performance of models trained on TRACE.

Figure 3: Performance concerning different trainable parameter ratios. The solid lines mark the performance of HFT with various ratios and the dashed lines mark the FFT baseline.

***Remark.*** *HFT is naturally suitable for scenarios with continual fine-tuning, and (almost all) methods with FFT can be further improved by assembling HFT, highlighting the plug-and-play feature.*

### 4.3 IMPACT OF PARAMETER SELECTION

HFT heuristically selects parameters to be tuned or frozen. We hope to reveal the impact of parameter selection from parameter radio and selection strategy, to discuss the universality of the methodology.

**Impact of Trainable Parameter Ratio.** Firstly, we traverse the radio of parameters to be fine-tuned at a granularity of $\sim$10% and evaluate the impact in both single-round and multi-round fine-tuning scenarios. *From Figure 3, we observe that most of the results with only updating partial parameters are superior to FFT, and the performance is quite satisfactory when the trainable parameter radio is around 50%.* In SFT, the performance of basic knowledge shows a clear downward trend with the increase of parameter ratio, while the general abilities slowly rise, which allows updating half or less of the parameters to have good performance. Meanwhile, when selecting half of the parameters during continual learning, the model reaches a balance of abilities between each round of tasks, resulting in a more robust training procedure and optimal performance. This observation again confirms the early conjecture about catastrophic forgetting, especially in continual learning, it is necessary to freeze a portion of parameters in each round to preserve the capabilities of the previous models. Not only that, we also find that fixing partial parameters gradually improves training efficiency (see Table 13), and HFT could shorten the training time by 30% in standard SFT.

**Impact of Selection Strategy.** Next, we consider other possible strategies for selecting half of the parameters: (1) **Model-level**. It arbitrarily chooses half the number of parameter matrices, which may prevent the parameter ratio from accurately reaching 50%. (2) **Layer-level**. It selects all parameters of a layer every other layer. (3) **Category-level**. It selects based on parameter categories, which is the default strategy used in this paper, and ensures the accurate selection

Table 4: Different strategies for selecting half of the parameters on TRACE.

|  | OP | BWT |
|---|---|---|
| SeqFT (FFT) | 45.7 | -10.2% |
| SeqFT (Model-level HFT) | 46.9 (+1.2) | -9.2% (+1.0%) |
| SeqFT (Layer-level HFT) | 47.9 (+2.2) | -8.3% (+1.9%) |
| SeqFT (Category-level HFT) | **51.3** (+5.6) | **-5.6%** (+4.6%) |

of 50% of the parameters. Table 4 reports the results of performing HFT on TRACE with sequential fine-tuning (SeqFT). *The first noteworthy phenomenon is that all three selection strategies outperform the standard FFT*, which once again confirms the motivation that freezing some parameters helps balance the old and new abilities in continual fine-tuning. *Moreover, the category-level selection wins the best performance*, we attribute it to the fine-grained strategy that maximizes the interaction between updated and non-updated parameters. From the perspective of model merging, it minimizes the damage to ready-made capabilities when performing a 50% dropout on the task vector, thereby providing greater possibilities for learning new tasks based on existing knowledge.

***Remark.*** *HFT is robust and insensitive to parameter selection, and selecting approximately 50% of the parameters with a reasonable selection strategy could achieve acceptable improvements.*

Table 5: General abilities and basic knowledge performance of HFT models fine-tuned on TÜLU V2 without `embedding` (E) and `lm_head` (H) layers. Note that the subscript indicates the proportion of selected parameters of `transformer` layers.

| | MMLU | GSM 8K | BBH | TyDi QA | Truthful QA | Human Eval | Natural Questions | Trivia QA | Hotpot QA | Overall |
|---|---|---|---|---|---|---|---|---|---|---|
| HFT$_{38.9\%}$ (update E, H) | 49.9 | 26.0 | 44.6 | 52.3 | 45.0 | 33.2 | 6.3 | 24.0 | 14.1 | 32.8 |
| HFT$_{50.0\%}$ (update E, H) | 50.8 | 30.5 | 43.6 | 52.3 | 45.4 | 34.6 | 6.2 | 32.8 | 15.4 | **34.6** |
| HFT$_{61.1\%}$ (update E, H) | 49.0 | 29.5 | 42.7 | 50.6 | 49.6 | 35.4 | 6.6 | 31.3 | 16.1 | 34.5 |
| HFT$_{50.0\%}$ (freeze E, H) | 51.4 | 29.0 | 45.0 | 50.5 | 45.2 | 35.0 | 3.2 | 24.1 | 13.7 | 33.0 |

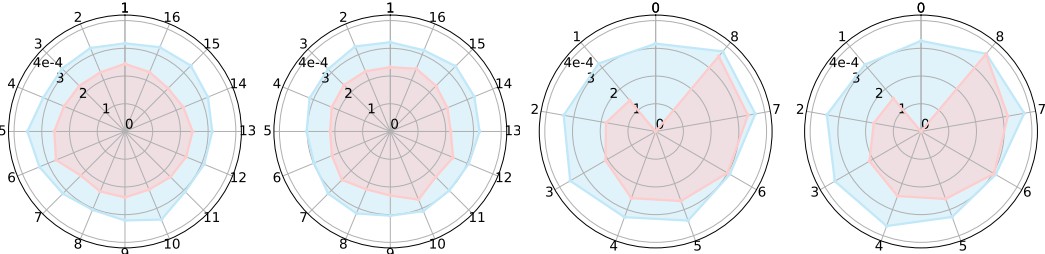

(a) Variations on SAN in various transformer blocks. (b) Variations on FFN in various transformer blocks. (c) Variations on SAN with various selected times. (d) Variations on FFN with various selected times.

Figure 4: Parameters variations of the last round model fine-tuned on TRACE relative to the starting point LLAMA 2-CHAT-7B. The outer blue circle indicates FFT and the inner red circle indicates HFT.

## 5 DISCUSSION

In this section, we further discuss the parameter changes in the fine-tuning process to deepen the understanding of HFT. We review the influence of `embedding` and `lm_head` layers, and visualize the parameter variations during successive training.

**Revisit the `Embedding` and `LM_head` Layers.**

HFT defaults to updating the `embedding` and `lm_head` layers. Here, we aim to explore the roles of these two input and output layers. Specifically, we freeze them while maintaining the same selection strategy and report results in supervised fine-tuning and continual learning. Since freezing the `embedding` and `lm_head` layers slightly reduces trainable parameters, we also include two models with similar parameter ratios that only freeze the parameters in `transformer` layers, to mitigate the impact of parameter ratio. As shown in Table 5, freezing these two layers leads to a substantial decline in knowledge-intensive

Table 6: OP and BWT scores of HFT models fine-tuned on TRACE without `embedding` and `lm_head` layers.

| | OP | BWT |
|---|---|---|
| HFT$_{38.9\%}$ (update E, H) | 49.6 | -5.6% |
| HFT$_{50.0\%}$ (update E, H) | **51.3** | -5.6% |
| HFT$_{61.1\%}$ (update E, H) | 49.9 | -5.6% |
| HFT$_{50.0\%}$ (freeze E, H) | 46.1 | **-2.2%** |

benchmarks, especially for QA-related tasks. Experimental results in Table 6 witness another phenomenon, where forgetting metric BWT significantly increases while the learning metric OP faces a cliff-like decrease. Detailed results in Appendix A.3.9 reveal that there is a substantial decline in the performance of ScienceQA. *To this extent, a preliminary conjecture emerges that the* `embedding` *and* `lm_head` *store information are highly relevant to world knowledge, so it is crucial to update them during the fine-tuning process.*

**Parameters Variation Analysis.** To intuitively perceive the difference in model parameters between HFT and FFT, we visualize parameter variations of fine-tuned models relative to the initial model (LLAMA 2-CHAT-7B) during continual learning on TRACE. On the one hand, we group two adjacent layers and calculate the average variation of self-attention and feed-forward blocks, where average variation refers to the average of all matrix differences in the block of two models. On the other hand, based on the selected number of times in these eight rounds of fine-tuning, we compare the average

variation of each block with FFT. Figure 4 shows variations from the perspective of the transformer block and selected time, respectively. Interestingly, we find that: (1) The parameter variation of each layer using HFT is fainter than those using FFT. (2) There is no significant difference in parameter variation between shallow and deep transformer layers, which is consistent in both fine-tuning settings. (3) The deviation from pre-trained parameters increases linearly with the time of selection, and the variations of parameters selected eight times are very similar to FFT. Therefore, the excessive offset of task vectors may not necessarily lead to an improvement in downstream performance but result in forgetting existing capabilities. *HFT seeks subtle balance by pulling back the task vector, alleviating catastrophic forgetting when learning subsequent tasks.*

## 6 RELATED WORK

**Sparse Fine-Tuning.** With the continuous increase in the number of language model parameters, sparse fine-tuning (a.k.a. parameter-efficient fine-tuning (PEFT)) offers an effective solution by reducing trainable parameters while achieving comparable performance to FFT (Fu et al., 2022; Ding et al., 2023; Han et al., 2024). Adapter (Houlsby et al., 2019; Mahabadi et al., 2021; Zhang et al., 2023a) and LoRA (Hu et al., 2022; Dou et al., 2023; Dettmers et al., 2023), the two most famous kinds of work, freeze the initial model weight and inject an adapter or a trainable rank decomposition matrices into each layer. However, these approaches change the model architecture and therefore require customized deployment. Keeping the architecture unchanged, DiffPruning (Guo et al., 2021) learns a sparse diff vector for each task, enabling PEFT to scale well with new tasks. BitFit (Zaken et al., 2021) only fine-tunes the bias terms of BERT and achieves considerably good performance. Unfortunately, these methods designed for specific tasks or networks (e.g., bias) are unsuitable for modern general-purpose large-scale models. From the perspective of low GPU memory overhead, BAdam (Luo et al., 2024) randomly divides the entire parameter into multiple blocks and updates each block sequentially, LISA (Pan et al., 2024) changes the granularity of blocks at the layer level. Besides, Mixout (Lee et al., 2020) resets a portion of neurons to a pre-trained state in each training step. In this way, all parameters in BAdam, LISA, and Mixout are updated, which is different from HFT and not conducive to continual learning.

**Continual Learning.** Continual learning aims to develop learning algorithms that can accumulate knowledge on non-stationary data, and vanilla FFT has been proven to lead to severe catastrophic forgetting issues when adapting to incoming streaming tasks (Luo et al., 2023; Wang et al., 2024). To address this issue, experience replay (Rolnick et al., 2019; Peng et al., 2024) is a widely used technique that incorporates a portion of data from previous rounds into the current training process. Regularization-based models (Kirkpatrick et al., 2017; Lopez-Paz & Ranzato, 2017) introduce additional terms in the loss function to penalize changes in crucial weights. Parameter-allocation approaches (Li et al., 2019; Gurbuz & Dovrolis, 2022) feature an isolated parameter subspace dedicated to each task throughout the network. When LLMs enter the era of billions of parameters, researchers prefer to use progressive prompts (Razdaibiedina et al., 2023) or PEFT (Dou et al., 2023; Wu et al., 2024a) to tune a powerful general backbone for specific tasks or domains (Wu et al., 2024b). Instead of introducing auxiliary modules or losses, HFT explores a new direction based on the characteristics of LLMs, proving that random parameter selection is sufficient to achieve passable performance and has the potential to become a successor to FFT.

## 7 CONCLUSION

In this paper, we observe that rolling back half of the fine-tuned parameters to the pre-trained state may recover partial knowledge of the startup model while holding the performance of downstream tasks. Taking inspiration from this observation, we propose Half Fine-tuning (HFT), which adopts a category-level strategy to select half of the parameters for updating in each training round, and the remaining parameters are expected to maintain the learned knowledge. Extensive experiments on supervised fine-tuning, direct preference optimization, and continual learning scenarios demonstrate the effectiveness of HFT. It not only alleviates the catastrophic forgetting in preceding capabilities but also achieves comparable or even superior performance than FFT in downstream tasks. Further analysis shows that HFT is robust to selection strategies and selected parameter numbers. Last but not least, HFT does not change the model architecture, making it easy to implement and scale, especially under successive fine-tuning scenarios.

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

# A APPENDIX

## A.1 BENEFITS AND LIMITATIONS

Half Fine-Tuning (HFT) achieves a balanced performance in general abilities and basic knowledge benchmarks. It outperforms the Full Fine-Tuning (FFT) strategy while saving approximately 30% of training time, and is scalable for scenarios with continual fine-tuning. In contrast, the widely used Sparse Fine-Tuning methods such as LoRA fall short of HFT in overall performance, and in more challenging scenarios like continual fine-tuning, these methods fail and lead to performance collapses. *We believe that HFT has the potential to become a successor to FFT in nearly all scenarios due to its superior performance and faster training speed.* Nonetheless, there are still some limitations to this paper. Firstly, due to computational resource constraints, we experiment with the most representative open-source models LLAMA 2-7B and LLAMA 2-13B, without scaling to larger or other family models. Secondly, we validate HFT on the standard dense transformer architecture, while other architectures such as Mixture-of-Experts (MoE) are not discussed in this paper. We believe that HFT is sufficient to adapt to other architectures and models, which warrants further research and exploration. In the future, we will strive to explore the potential of HFT in a wider range and diverse architecture models, while also refining selection methods to further improve performance.

## A.2 EXPERIMENTAL SETUP

### A.2.1 DATASETS

To validate the performance of supervised fine-tuning, we choose TÜLU V2 (Ivison et al., 2023) which is a combination of high-quality open resources, including datasets (1) created by researchers from existing NLP datasets (e.g. SuperNI (Wang et al., 2022)), (2) written by humans (e.g. Dolly (Conover et al., 2023) and Open Assistant (Köpf et al., 2023)), (3) generated by LLMs (e.g. Self-Instruct (Wang et al., 2023b), Alpaca (Taori et al., 2023) and Baize (Xu et al., 2023)), (4) comprised of user-shared prompts accompanied by model-generated completions (e.g. ShareGPT (Chiang et al., 2023)), and (5) developed for specific abilities (e.g. CoT (Wei et al., 2022) for chain-of-thought and Code-Alpaca (Chaudhary, 2023) for code generation). To examine the capacity for reinstating a fraction of impaired capabilities while adhering to human preferences, we utilize **UltraFeedback** (Cui et al., 2023) which is a large-scale, high-quality, and diversified preference dataset. For continual learning, we select **TRACE** (Wang et al., 2023a), a novel benchmark designed for continual learning (CL) in LLMs, to evaluate catastrophic forgetting in standard CL settings. TRACE consists of 8 distinct datasets spanning challenging tasks, domain-specific tasks, multilingual capabilities, code generation, and mathematical reasoning.

### A.2.2 EVALUATION METRICS

**Supervised Fine-Tuning and Direct Preference Optimization.** To validate the effectiveness of our method, we employ general abilities and basic knowledge benchmarks to assess the performance in learning new tasks and preserving the original capabilities, respectively. Specifically, for the *general abilities benchmarks*, we include the following evaluation sets to test various abilities. (1) **Factual knowledge**: To assess the LLMs' factual knowledge, we employ the Massive Multitask Language Understanding dataset (**MMLU**) (Hendrycks et al., 2021). MMLU comprises a collection of questions across 57 subjects from elementary to professional difficulty levels. We report the 5-shot accuracy based on answer perplexity. (2) **Reasoning**: We utilize the test split of the Grade School Math (**GSM8K**) dataset (Cobbe et al., 2021) and Big-Bench-Hard (**BBH**) (Suzgun et al., 2023) to evaluate the reasoning abilities. We report the 8-shot accuracy and the exact match (EM) rates for GSM8K and BBH, respectively. (3) **Multilingualism**: To evaluate multilingual capabilities, we employ **TyDiQA** (Clark et al., 2020), a multilingual question-answering benchmark that covers 11 typologically diverse languages. We adopt the gold-passage setup, where a passage containing the reference answer is provided, and report the F1 score. (4) **Coding**: To evaluate the LLMs' ability to generate functionally correct programs from docstrings, we utilize **HumanEval** (Chen et al., 2021) and report the pass@10 performance. (5) **Truthful**: We incorporate **TruthfulQA** (Lin et al., 2022) to assess the ability to avoid generating known falsehoods resulting from misconceptions or false beliefs while providing informative responses. (6) **Conversation**: We use **AlpacaEval 2.0** (Li et al., 2023) to evaluate the instruction-following abilities. AlpacaEval is an LLM-based automatic evaluation metric.

**Algorithm 1:** Algorithm of HFT with Category-Leval Parameter Selection

---

**Input:** Pre-trained model $\theta_0$

Initialize sequential training task $\mathcal{T}$ with data $\mathcal{D}_t$, feed-forward block container `FFNs=[]`, self-attention block container `SANs=[]`, and layernorm block container `LNs=[]`.

**for** $t = 1$ to $|\mathcal{T}|$ **do**
    *// Set all parameters to retain gradients before each fine-tuning stage*
    **foreach** *param in* $\theta_{t-1}$ **do**
        └ `param.requires_grad = True`
    *// Omit the embedding and lm_head layer*
    `mark_layers = random.sample(transformer_layers,`
     `len(transformer_layers)//2)`
    **foreach** *layer in transformer_layers* **do**
        **foreach** *param in layer* **do**
            **if** *param **belongs to** FFN block* **then**
                └ `FFNs.append(param)`
            **else if** *param **belongs to** SAN block* **then**
                └ `SANs.append(param)`
            **else**
                └ `LNs.append(param)`
        *// For FFNs with an odd number of parameters in one layer, the number of selected parameters in half of the layers is rounded up, while the other half is rounded down.*
        **if** *layer in mark_layers* **then**
        └ `freeze_ffn = random.sample(FFNs, `$\lceil$`len(FFNs)/2`$\rceil$`)`
        **else**
        └ `freeze_ffn = random.sample(FFNs, `$\lfloor$`len(FFNs)/2`$\rfloor$`)`
        `freeze_san = random.sample(SANs, len(SANs)//2)`
        `freeze_ln = random.sample(LNs, len(LNs)//2)`
        **foreach** *param in freeze_ffn, freeze_san **and** freeze_ln* **do**
        └ `param.requires_grad = False`
    └ Set `FFNs`, `SANs` and `LNs` to `[]`
    └ Model training process on with dataset $\mathcal{D}_t$

**Output:** Fine-tuned model $\theta_{|\mathcal{T}|}$

---

In this paper, we calculate the win rates against the `GPT-4-preview-1106`. We include the following three datasets for *basic knowledge benchmarks* to validate the basic knowledge preserved in LLMs: (1) **NaturalQuestion** (Kwiatkowski et al., 2019), (2) **TriviaQA** (Han et al., 2019), and (3) **HotpotQA** (Yang et al., 2018).

**Continual Learning.** For continual learning evaluations, following (Wang et al., 2023a), we use Overall Performance (OP) and Backward Transfer (BWT) scores as the main metrics in CL settings. In terms of the formula, after incrementally learning the $t$-th task, the performance on the $i$-th task (where $i \leq t$) is denoted as $S_{t,i}$. The OP and BWT scores can be calculated as

$$\text{OP}_t = \frac{1}{t}\sum_{i=1}^{t} S_{t,i}, \quad \text{BWT}_t = \frac{1}{t}\sum_{i=1}^{t-1}\left(S_{t,i} - S_{i,i}\right). \tag{5}$$

We utilize accuracy as the primary evaluation metric for C-STANCE, FOMC, ScienceQA, NumGLUE-cm, and NumGLUE-ds. In the case of Py150, we employ similarity as the evaluation metric. Moreover, for the evaluation of MeetingBank and 20Minuten, we employ the ROUGE-L metric.

### A.2.3 IMPLEMENTATION DETAILS

Following (Ivison et al., 2023), in the SFT phase on TÜLU V2, we adopt a linear-decreasing learning rate of 2e-5 with a 0.3 warmup ratio and train for 2 epochs. For the human preference alignment phase on UltraFeedback, we use direct preference optimization (Rafailov et al., 2023) to align the fine-tuned LLMs on TÜLU V2. We use a learning rate of 5e-7 and a global batch size of 32. Due to the context length of 4096 used during LLAMA 2 pre-training, as referenced in the (Ivison et al., 2023) code repository issues, we set a maximum sequence length of 4096 during the SFT stage.

Table 7: General abilities and basic knowledge results of LLAMA 2-7B, the well-aligned model LLAMA 2-CHAT-7B, and our proposed three half-reset approaches.

| | LLAMA 2-7B | LLAMA 2-CHAT-7B | Model-level Half-Reset | Layer-level Half-Reset | Category-level Half-Reset |
|---|---|---|---|---|---|
| MMLU (EM, 0-shot) | 41.6 | **47.0** | 46.2 | 45.8 | 46.7 |
| GSM (ACC, 8-shot) | 12.0 | **26.0** | 8.0 | 22.0 | 24.0 |
| BBH (EM, 0-shot) | 39.9 | 39.2 | **41.0** | 39.5 | 37.7 |
| TyDiQA (F1, 1-shot) | **48.4** | 43.6 | 46.3 | 44.2 | 44.9 |
| TruthfulQA (MC2, 0-shot) | 38.5 | **46.0** | 41.7 | 43.1 | 41.7 |
| HumanEval (Pass@10) | 26.2 | 23.9 | **26.8** | 25.0 | 22.0 |
| *Overall (General Ability)* | 34.4 | **37.6** | 35.0 | 36.6 | 36.2 |
| NaturalQuestion (EM, 0-shot) | **12.9** | 7.2 | 8.2 | 11.2 | 10.9 |
| TriviaQA (EM, 0-shot) | **40.2** | 3.3 | 18.3 | 21.3 | 21.3 |
| HotpotQA (EM, 0-shot) | **15.6** | 6.6 | 7.4 | 9.9 | 9.0 |
| *Overall (World Knowledge)* | **22.9** | 5.7 | 11.3 | 12.4 | 13.7 |
| *Overall* | **30.6** | 27.0 | 27.1 | 28.5 | 28.7 |

However, due to hardware resource limitations, the maximum sequence length is reduced to 1024 during the DPO stage under LLAMA 2-13B. During the continual learning phase, following (Wang et al., 2023a), we employ a fixed learning rate of 1e-5 and fine-tune the eight sub-datasets for different numbers of epochs: 5, 3, 7, 5, 3, 5, 5, and 7 epochs, respectively. The global batch size for both stages is set to 128. All our experiments are conducted on one machine equipped with 8x80G Nvidia A100. Algorithm 1 introduce the detailed implementations of our proposed fine-grained selecting approach of HFT. Additionally, to evaluate the SFT and DPO models, we employ a chat format, using specialized tokens `<|user|>` and `<|assistant|>` to mark user utterances and target assistant responses, respectively. However, we use a standard language format for HumanEval and the basic knowledge benchmarks when evaluating pre-trained models.

## A.3 ADDITIONAL EXPERIMENTS

### A.3.1 DETAILED RESULTS OF PILOT EXPERIMENTS

Table 7 presents the detailed results of pilot experiments conducted in Section 2. We also compare two additional model-level and layer-level parameter selection methods here. The results indicate that the category-level selection approach achieves the highest overall performance, consistent with the follow-up training setting conclusion.

### A.3.2 MORE BASELINES OF INSTRUCTION TUNING

We introduce two extra groups of methods to illustrate the effectiveness of HFT. Specifically, we compare four sparse fine-tuning methods, LoRA (Hu et al., 2022), QLoRA (Dettmers et al., 2023), AdaLoRA (Zhang et al., 2023b), P-Tuning (Liu et al., 2022), and Mixout (Lee et al., 2020) as well as three model merging methods, Average merging, TIES merging (Yadav et al., 2023), and DARE (Yu et al., 2023). The experimental results are shown in Table 8, demonstrating that the HFT method achieves the best trade-off in both general abilities and basic knowledge benchmarks. The sparse fine-tuning methods preserve more basic knowledge but suffer more performance degradation in the general abilities evaluation, which is consistent with the previous conclusion that LoRA learns less and forgets less (Biderman et al., 2024). On the other hand, the model merging methods, in general, also perform worse than HFT. Additionally, model merging methods require FFT training followed by task vector pruning, making them more complex and time-consuming due to the two-stage process.

### A.3.3 DIRECT PREFERENCE OPTIMIZATION WITH HFT-BASED MODELS

In Section 4.1, we initialize our DPO process with the FFT model. In this section, we investigate the performance of the DPO process when initialized with the HFT model. The experimental results are presented in Table 9. We observe that while the DPO process on the HFT model performs better

Table 8: General abilities and basic knowledge performance of more baselines. In model merging baselines, P, S and D refer to Pre-trained, SFT and DPO models, respectively.

| | MMLU | GSM 8K | BBH | TyDi QA | Truthful QA | Human Eval | Natural Question | Trivia QA | Hotpot QA | Overall |
|---|---|---|---|---|---|---|---|---|---|---|
| *Sparse Fine-tuning Baselines* | | | | | | | | | | |
| LoRA | 46.8 | 18.0 | 39.5 | 51.7 | 44.8 | 27.3 | 12.7 | 36.2 | **17.8** | 32.8 |
| QLoRA | 38.0 | 2.5 | 37.2 | 15.0 | 40.6 | 24.0 | 12.7 | **43.2** | 15.5 | 25.4 |
| AdaLoRA | 47.2 | 19.5 | 39.1 | 51.9 | 44.4 | 30.2 | 12.3 | 37.5 | 16.9 | 33.2 |
| P-tuning | 44.7 | 16.5 | 36.9 | 50.2 | 43.6 | 26.5 | 12.8 | 40.9 | 17.3 | 32.2 |
| Mixout | 48.1 | 24.5 | 41.0 | 49.8 | 42.3 | 33.7 | 4.5 | 28.2 | 15.5 | 32.0 |
| *Model Merging Baselines* | | | | | | | | | | |
| TIES (P+S) | 47.8 | 25.5 | 40.2 | 50.1 | 43.3 | 30.2 | 5.5 | 31.7 | 14.4 | 32.1 |
| DARE (P+S) | 49.2 | 28.5 | 42.9 | **53.0** | 44.4 | 32.8 | 6.1 | 30.7 | 15.1 | 33.6 |
| TIES (S+D) | 39.6 | 1.5 | 39.7 | 16.1 | 38.4 | 23.3 | **12.9** | 40.2 | 15.6 | 25.3 |
| DARE (S+D) | 45.8 | 16.5 | 40.4 | 50.0 | 42.7 | 27.6 | 5.8 | 32.7 | 14.1 | 30.6 |
| Average (S+D) | 49.0 | 22.0 | **45.1** | 52.8 | 42.5 | 32.6 | 7.5 | 35.6 | 14.0 | 33.5 |
| HFT (S) | **50.8** | **30.5** | 43.6 | 52.3 | **45.4** | **34.6** | 6.2 | 32.8 | 15.4 | **34.6** |

Table 9: General abilities and basic knowledge performance of DPO stage (with HFT), which is initialized with HFT-based SFT models fine-tuned on TÜLU V2.

| | MMLU | GSM 8K | BBH | TyDi QA | Truthful QA | Human Eval | Natural Question | Trivia QA | Hotpot QA | Overall |
|---|---|---|---|---|---|---|---|---|---|---|
| DPO (FFT-based, 7b) | 48.8 | 25.5 | 42.8 | 51.1 | 45.5 | 36.7 | 1.9 | 22.9 | 12.8 | **32.0** |
| DPO (HFT-based, 7b) | 50.7 | 30.5 | 42.8 | 43.9 | 49.8 | 35.1 | 1.0 | 20.4 | 5.9 | 31.1 |
| DPO (FFT-based, 13b) | 51.8 | 48.5 | 49.9 | 52.9 | 45.3 | 41.0 | 0.2 | 5.5 | 3.0 | 33.1 |
| DPO (HFT-based, 13b) | 55.0 | 45.5 | 51.4 | 53.2 | 49.5 | 42.9 | 0.3 | 4.9 | 4.7 | **34.2** |

in certain general abilities, such as TruthfulQA, it experiences minor losses in overall performance under LLAMA 2-7B. However, the situation is reversed in LLAMA 2-13B, where the DPO deployed on the HFT model outperforms the FFT-initialized DPO. Nonetheless, DPO equipped with HFT tends to improve performance compared to DPO with FFT consistently.

### A.3.4    GENERAL ABILITIES AND BASIC KNOWLEDGE OF CONTINUAL FINE-TUNED MODELS

We also evaluate the models mentioned in Section 4.2 on general abilities and basic knowledge benchmarks. The experimental results are presented in Table 10. We observe that after 8 rounds of fine-tuning on consecutive tasks, the models fine-tuned with the HFT method consistently outperform the FFT models in terms of overall performance. This further confirms the effectiveness of HFT in preserving the original capabilities of the model and mitigating catastrophic forgetting. Furthermore, although LoRA preserves more layer parameters unchanged, it still performs worse compared to HFT. We believe this may be attributed to the low-rank decomposition resulting in a limited number of trainable parameters. Merging the LoRA weights back into the original model could potentially disrupt the original parameter space to a greater extent.

### A.3.5    THE IMPACT OF LEARNING RATES

To validate whether our approach indeed leverages the frozen parameters to mitigate the catastrophic forgetting, rather than being equivalent to the effects brought about by a reduced learning rate, we compare the half learning rate and the cosine learning rate schedule to demonstrate further that the way HFT alleviates forgetting is not depending on learning rate but is indeed due to the role played by the frozen parameters. As shown in Tabel 11, we observe that upon halving the learning rate, the overall performance declines, with no significant recovery in the performance on world knowledge, thereby underscoring the capability of HFT in mitigating catastrophic forgetting. Moreover, under

Table 10: General abilities and basic knowledge performance of the final round models fine-tuned on TRACE. We compare four different fine-tuning methods and our HFT approach start from LLAMA 2-CHAT-7B and LLAMA 2-CHAT-13B.

| | MMLU | GSM 8K | BBH | TyDi QA | Truthful QA | Human Eval | Natural Question | Trivia QA | Hotpot QA | Overall |
|---|---|---|---|---|---|---|---|---|---|---|
| SeqFT-7b | 35.5 | 3.0 | 24.3 | 39.1 | 42.7 | 0.3 | 10.0 | 23.9 | 14.0 | 21.4 |
| GEM-7b | 40.1 | 3.5 | 17.0 | 33.4 | 41.4 | 2.2 | 10.0 | 19.6 | 14.0 | 20.1 |
| Replay-7b | 45.9 | 4.5 | 35.2 | 41.6 | 39.6 | 8.5 | 11.6 | 36.1 | 14.2 | 26.4 |
| LoraSeqFT-7b | 43.3 | 11.0 | 30.7 | 35.5 | 41.7 | 8.8 | 8.7 | 24.7 | 13.4 | 24.2 |
| SeqFT-7b (H) | 44.1 | 3.5 | 30.8 | 41.1 | 41.8 | 1.6 | 11.3 | 38.9 | 14.4 | **25.3** (+3.9) |
| GEM-7b (H) | 45.1 | 5.0 | 32.3 | 34.9 | 43.0 | 2.7 | 10.4 | 35.9 | 13.7 | **24.8** (+4.7) |
| Replay-7b (H) | 47.9 | 11.0 | 38.8 | 42.6 | 42.5 | 12.7 | 10.7 | 38.4 | 12.9 | **28.6** (+2.2) |
| SeqFT-13b | 39.7 | 5.0 | 27.9 | 41.0 | 41.4 | 0.0 | 12.7 | 44.3 | 16.3 | 25.4 |
| Replay-13b | 49.0 | 3.5 | 40.1 | 37.7 | 43.1 | 12.0 | 12.5 | 6.7 | 13.3 | 24.2 |
| GEM-13b | 47.2 | 4.0 | 37.6 | 36.3 | 43.0 | 10.0 | 10.8 | 10.2 | 12.1 | 23.5 |
| LoraSeqFT-13b | 43.3 | 15.0 | 42.4 | 43.1 | 40.5 | 18.2 | 10.6 | 37.6 | 16.2 | 29.7 |
| SeqFT-13b (H) | 50.0 | 7.0 | 46.3 | 47.2 | 41.4 | 11.2 | 14.7 | 50.6 | 18.7 | **31.9** (+6.5) |
| GEM-13b (H) | 49.9 | 9.5 | 46.5 | 38.2 | 45.1 | 18.9 | 9.8 | 39.7 | 14.2 | **30.2** (+6.7) |
| Replay-13b (H) | 50.0 | 10.5 | 47.1 | 39.6 | 45.8 | 20.1 | 10.1 | 41.1 | 14.0 | **30.9** (+2.3) |

Table 11: General abilities and basic knowledge of LLAMA 2 7B based on different learning rates.

| | FFT (linear,1e-5) | FFT (linear,2e-5) | HFT (linear,2e-5) | FFT (cosine,2e-5) | HFT (cosine,2e-5) |
|---|---|---|---|---|---|
| MMLU (EM, 0-shot) | 49.2 | 48.5 | **50.8** | 47.8 | 50.6 |
| GSM (ACC, 8-shot) | 24.5 | 25.0 | 30.5 | 25.5 | **31.5** |
| BBH (EM, 0-shot) | 41.8 | 42.2 | 43.6 | 42.2 | **44.4** |
| TyDiQA (F1, 1-shot) | 51.5 | 51.2 | 52.3 | 51.2 | **52.8** |
| TruthfulQA (MC2, 0-shot) | 40.2 | 41.7 | 45.4 | 42.6 | **46.4** |
| HumanEval (Pass@10) | 36.0 | **36.9** | 34.6 | 34.3 | 33.7 |
| *Overall (General Ability)* | 40.4 | 41.0 | 42.9 | 40.6 | **43.2** |
| NaturalQuestion (EM, 0-shot) | 4.9 | 3.2 | 6.2 | 3.5 | **6.4** |
| TriviaQA (EM, 0-shot) | 22.7 | 26.4 | 32.8 | 27.6 | **33.6** |
| HotpotQA (EM, 0-shot) | 13.4 | 14.5 | **15.4** | 13.1 | 14.7 |
| *Overall (World Knowledge)* | 13.7 | 14.7 | 18.1 | 14.7 | **18.2** |
| *Overall* | 31.5 | 32.2 | 34.6 | 32.0 | **34.9** |

the cosine learning rate schedule, HFT still outperforms FFT, which also demonstrates the robustness of HFT to variations in the learning rate.

### A.3.6 THE IMPACT OF RANDOMNESS

Here, we discuss a series of factors related to the randomness of HFT, including different trainable parameter ratios and selection methods. Note that in the continual learning setting, we randomly select trainable parameters for each fine-tuning process, with a total of 8 random selections. The significant performance improvement of HFT over FFT indicates that it is not sensitive to fine-grained parameter selection. For all that, we also supplement a randomness experiment under the instruction tuning setting with 5 different random seeds (i.e. parameter selections). As shown in Table 12, among these 5 trials, HFT exhibits minimal variations and a stable lead relative to FFT, demonstrating its robustness again.

### A.3.7 EFFICIENCY ANALYSIS

We conduct a comparison of the runtime costs for different ratios of trainable parameters. Specifically, we fine-tuned LLAMA 2-7B on TÜLU V2 and record the total duration from the start to the end of

Table 12: General abilities and basic knowledge of LLAMA 2 7B based on different random seeds.

| | HFT (seed 1) | HFT (seed 2) | HFT (seed 3) | HFT (seed 4) | HFT (seed 5) |
|---|---|---|---|---|---|
| MMLU (EM, 0-shot) | 50.8 | 49.9 | 50.2 | **51.2** | 50.5 |
| GSM (ACC, 8-shot) | 30.5 | **31.0** | 30.5 | 28.5 | 29.5 |
| BBH (EM, 0-shot) | 43.6 | 43.2 | 42.9 | 43.4 | **44.1** |
| TyDiQA (F1, 1-shot) | 52.3 | 52.3 | **53.2** | 52.8 | 51.7 |
| TruthfulQA (MC2, 0-shot) | 45.4 | **45.7** | 44.7 | 45.2 | 44.9 |
| HumanEval (Pass@10) | 34.6 | 35.1 | 34.8 | 34.7 | **35.2** |
| *Overall (General Ability)* | **42.9** | **42.9** | 42.7 | 42.6 | 42.7 |
| NaturalQuestion (EM, 0-shot) | 6.2 | 6.1 | 5.9 | 6.1 | **6.4** |
| TriviaQA (EM, 0-shot) | 32.8 | 31.9 | **33.4** | 33.1 | 33.0 |
| HotpotQA (EM, 0-shot) | 15.4 | 15.4 | **15.6** | 14.9 | **15.6** |
| *Overall (World Knowledge)* | 18.1 | 17.8 | **18.3** | 18.0 | **18.3** |
| *Overall* | **34.6** | 34.5 | **34.6** | 34.4 | **34.6** |

Table 13: Efficiency analysis among different ratios of trainable parameters, in which FFT as a reference value and underline marks HFT proposed in this paper.

| # Trainable Parameters (%) | 8.3 | 22.3 | 30.6 | 38.9 | 50.0 | 61.1 | 69.4 | 77.7 | 91.7 | 100 |
|---|---|---|---|---|---|---|---|---|---|---|
| Runtime (%) | 48.0 | 52.2 | 56.4 | 64.0 | 68.5 | 72.5 | 85.1 | 85.2 | 89.0 | 100 |
| Δ (%) | -52.0 | -47.8 | -43.6 | -36.0 | -31.5 | -27.5 | -14.9 | -14.8 | -11.0 | 0.0 |

the training program. The results in Table 13 demonstrate that, without specific optimization, all models with varying ratios of trainable parameters can reduce the training time. As expected, as the proportion of trainable parameters increases, the training duration also increases. Notably, our HFT method achieves a 31.5% reduction in training time, significantly decreasing the training cost for extremely large-scale instruction datasets.

### A.3.8   EVALUATION ON ALPACAEVAL

As shown in Table 14, we evaluate different models on AlpacaEval 2.0. The results indicate that our method is less effective than FFT on LLAMA 2-7B. However, a reversal occurs when the model size scales up to 13b, where our approach outperforms the FFT models comprehensively. This suggests that our method has greater potential on much larger-scale LLMs, as supported by the experimental results in Table 1, which show a larger improvement of HFT compared to FFT on LLAMA 2-13B compared to LLAMA 2-7B. Interestingly, the Half-Reset method performs well on LLAMA 2-13B but shows completely different results on LLAMA 2-7B. This suggests that simply resetting half of the parameters may not provide consistent performance since the model is trained on the full set of parameters.

Table 14: Evaluation results on AlpacaEval 2.0.

| Models | AlpacaEval 2.0 |
|---|---|
| LLAMA 2-7B-SFT | **6.96** |
| LLAMA 2-7B-SFT (R) | 2.98 |
| LLAMA 2-7B-SFT (H) | 5.59 |
| LLAMA 2-7B-DPO | **10.68** |
| LLAMA 2-7B-DPO (R) | 8.44 |
| LLAMA 2-7B-DPO (H) | 9.07 |
| LLAMA 2-13B-SFT | 8.32 |
| LLAMA 2-13B-SFT (R) | **11.93** |
| LLAMA 2-13B-SFT (H) | 10.43 |
| LLAMA 2-13B-DPO | 11.55 |
| LLAMA 2-13B-DPO (R) | **12.55** |
| LLAMA 2-13B-DPO (H) | 11.68 |

### A.3.9   DETAILED RESULTS OF REVISITING EMBEDDING AND LM_HEAD LAYERS

Table 15 details the results of freezing the input and output layers. Meanwhile, Table 16 and 17 show the detailed results of the two adjacent numbers of parameter settings on TRACE.

Table 15: Detailed results on TRACE with 50.0% trainable parameters while freezing `embedding` and `lm_head` layers.

| Task\Round | 1 | 2 | 3 | 4 | 5 | 6 | 7 | 8 |
|---|---|---|---|---|---|---|---|---|
| C-STANCE | 50.1 | 48.0 | 47.2 | 45.8 | 46.4 | 46.2 | 46.3 | 48.0 |
| FOMC | - | 69.0 | 66.1 | 65.7 | 65.7 | 64.7 | 63.9 | 66.9 |
| MeetingBank | - | - | 37.5 | 34.5 | 34.2 | 32.7 | 31.9 | 33.2 |
| Py150 | - | - | - | 51.2 | 50.3 | 49.8 | 49.2 | 50.8 |
| ScienceQA | - | - | - | - | 58.1 | 58.0 | 56.8 | 56.2 |
| NumGLUE-cm | - | - | - | - | - | 33.3 | 25.9 | 29.6 |
| NumGLUE-ds | - | - | - | - | - | - | 45.8 | 43.1 |
| 20Minuten | - | - | - | - | - | - | - | 40.6 |
| **OP** | **50.1** | **58.5** | **50.3** | **49.3** | **50.9** | **47.5** | **45.7** | **46.1** |
| **BWT** | - | - | - | - | - | - | - | **-2.2%** |

Table 16: Detailed results on TRACE with 38.9% trainable parameters while updating `embedding` and `lm_head` layers.

| Task\Round | 1 | 2 | 3 | 4 | 5 | 6 | 7 | 8 |
|---|---|---|---|---|---|---|---|---|
| C-STANCE | 49.2 | 43.7 | 43.2 | 44.2 | 44.2 | 44.4 | 43.7 | 45.1 |
| FOMC | - | 71.0 | 64.3 | 65.3 | 60.7 | 65.9 | 65.1 | 63.3 |
| MeetingBank | - | - | 46.9 | 37.7 | 35.4 | 39.0 | 38.5 | 36.9 |
| Py150 | - | - | - | 57.9 | 52.6 | 53.6 | 53.6 | 53.4 |
| ScienceQA | - | - | - | - | 85.7 | 77.5 | 71.8 | 74.8 |
| NumGLUE-cm | - | - | - | - | - | 33.3 | 29.6 | 33.3 |
| NumGLUE-ds | - | - | - | - | - | - | 56.6 | 48.9 |
| 20Minuten | - | - | - | - | - | - | - | 41.1 |
| **OP** | **49.2** | **57.4** | **51.5** | **51.3** | **55.7** | **52.3** | **51.3** | **49.6** |
| **BWT** | - | - | - | - | - | - | - | **-5.6%** |

Table 17: Detailed results on TRACE with 61.1% trainable parameters while updating `embedding` and `lm_head` layers.

| Task\Round | 1 | 2 | 3 | 4 | 5 | 6 | 7 | 8 |
|---|---|---|---|---|---|---|---|---|
| C-STANCE | 45.3 | 50.8 | 50.9 | 51.4 | 51.3 | 51.4 | 51.1 | 53.3 |
| FOMC | - | 72.8 | 63.7 | 65.7 | 6.3 | 68.3 | 69.0 | 67.9 |
| MeetingBank | - | - | 48.9 | 41.1 | 38.3 | 41.3 | 41.1 | 40.0 |
| Py150 | - | - | - | 57.3 | 50.3 | 52.8 | 52.9 | 52.9 |
| ScienceQA | - | - | - | - | 88.2 | 70.6 | 67.3 | 69.4 |
| NumGLUE-cm | - | - | - | - | - | 30.9 | 28.4 | 21.0 |
| NumGLUE-ds | - | - | - | - | - | - | 59.4 | 53.5 |
| 20Minuten | - | - | - | - | - | - | - | 40.8 |
| **OP** | **45.3** | **61.8** | **54.5** | **53.9** | **46.9** | **52.6** | **52.7** | **49.9** |
| **BWT** | - | - | - | - | - | - | - | **-5.6%** |

### A.3.10 DETAILED RESULTS OF DIFFERENT PARAMETER SELECTION STRATEGIES

Table 18 and 19 provide the detailed results on TRACE with model-level and layer-level parameter selection strategies mentioned in Section 4.3.

### A.3.11 DETAILED RESULTS OF TRACE

Table 20 to 33 show the detailed results of different models and approaches of each round during the continual learning on TRACE.

Table 18: Detailed results on TRACE with model-level parameter selection.

| Task\Round | 1 | 2 | 3 | 4 | 5 | 6 | 7 | 8 |
|---|---|---|---|---|---|---|---|---|
| C-STANCE | 49.3 | 49.1 | 48.8 | 50.2 | 50.0 | 48.9 | 48.1 | 49.2 |
| FOMC | - | 70.6 | 57.5 | 53.8 | 42.7 | 54.4 | 58.1 | 55.2 |
| MeetingBank | - | - | 48.9 | 37.8 | 36.5 | 38.2 | 37.3 | 38.9 |
| Py150 | - | - | - | 57.7 | 55.4 | 55.9 | 54.8 | 55.7 |
| ScienceQA | - | - | - | - | 87.7 | 59.8 | 54.2 | 56.4 |
| NumGLUE-cm | - | - | - | - | - | 38.3 | 22.2 | 25.9 |
| NumGLUE-ds | - | - | - | - | - | - | 55.7 | 53.5 |
| 20Minuten | - | - | - | - | - | - | - | 40.7 |
| **OP** | **49.3** | **59.9** | **51.7** | **49.9** | **54.5** | **49.3** | **47.2** | **46.9** |
| **BWT** | - | - | - | - | - | - | - | **-9.2%** |

Table 19: Detailed results on TRACE with layer-level parameter selection.

| Task\Round | 1 | 2 | 3 | 4 | 5 | 6 | 7 | 8 |
|---|---|---|---|---|---|---|---|---|
| C-STANCE | 50.8 | 41.4 | 44.6 | 46.5 | 47.5 | 48.6 | 48.2 | 49.0 |
| FOMC | - | 72.2 | 58.5 | 54.6 | 1.8 | 46.8 | 50.2 | 50.0 |
| MeetingBank | - | - | 47.1 | 34.7 | 34.5 | 37.2 | 38.6 | 37.1 |
| Py150 | - | - | - | 56.5 | 53.3 | 53.8 | 54.2 | 54.1 |
| ScienceQA | - | - | - | - | 88.5 | 84.4 | 76.2 | 77.5 |
| NumGLUE-cm | - | - | - | - | - | 35.8 | 28.4 | 21.0 |
| NumGLUE-ds | - | - | - | - | - | - | 57.2 | 52.9 |
| 20Minuten | - | - | - | - | - | - | - | 41.5 |
| **OP** | **50.8** | **56.8** | **50.1** | **48.1** | **45.1** | **51.1** | **50.4** | **47.9** |
| **BWT** | - | - | - | - | - | - | - | **-8.3%** |

Table 20: Detailed results on TRACE with SeqFT (start from LLAMA 2-CHAT-7B).

| Task\Round | 1 | 2 | 3 | 4 | 5 | 6 | 7 | 8 |
|---|---|---|---|---|---|---|---|---|
| C-STANCE | 48.5 | 49.7 | 48.5 | 48.3 | 6.7 | 47.4 | 47.2 | 48.7 |
| FOMC | - | 71.6 | 46.6 | 46.4 | 0.4 | 43.1 | 42.9 | 44.0 |
| MeetingBank | - | - | 49.0 | 39.9 | 40.8 | 37.6 | 34.5 | 37.9 |
| Py150 | - | - | - | 57.0 | 49.2 | 54.5 | 54.2 | 54.0 |
| ScienceQA | - | - | - | - | 89.1 | 71.5 | 44.6 | 60.6 |
| NumGLUE-cm | - | - | - | - | - | 30.9 | 24.7 | 25.9 |
| NumGLUE-ds | - | - | - | - | - | - | 59.4 | 52.6 |
| 20Minuten | - | - | - | - | - | - | - | 41.5 |
| **OP** | **48.5** | **60.7** | **48.0** | **47.9** | **37.2** | **47.5** | **43.9** | **45.7** |
| **BWT** | - | - | - | - | - | - | - | **-10.2%** |

Table 21: Detailed results on TRACE with SeqFT and HFT (start from LLAMA 2-CHAT-7B).

| Task\Round | 1 | 2 | 3 | 4 | 5 | 6 | 7 | 8 |
|---|---|---|---|---|---|---|---|---|
| C-STANCE | 49.4 | 47.6 | 45.6 | 46.4 | 47.8 | 49.5 | 49.1 | 49.3 |
| FOMC | - | 71.8 | 57.7 | 59.1 | 46.0 | 66.5 | 67.3 | 66.3 |
| MeetingBank | - | - | 47.4 | 39.1 | 31.2 | 38.6 | 38.4 | 35.7 |
| Py150 | - | - | - | 57.4 | 52.1 | 54.8 | 55.0 | 55.0 |
| ScienceQA | - | - | - | - | 87.4 | 82.1 | 77.6 | 75.3 |
| NumGLUE-cm | - | - | - | - | - | 42.0 | 30.9 | 32.1 |
| NumGLUE-ds | - | - | - | - | - | - | 58.5 | 55.1 |
| 20Minuten | - | - | - | - | - | - | - | 41.3 |
| **OP** | **49.4** | **59.7** | **50.2** | **50.5** | **52.9** | **55.6** | **53.8** | **51.3** |
| **BWT** | - | - | - | - | - | - | - | **-5.6%** |

Table 22: Detailed results on TRACE with GEM (start from LLAMA 2-CHAT-7B).

| Task\Round | 1 | 2 | 3 | 4 | 5 | 6 | 7 | 8 |
|---|---|---|---|---|---|---|---|---|
| C-STANCE | 50.0 | 48.9 | 48.4 | 47.7 | 13.0 | 46.5 | 45.7 | 48.1 |
| FOMC | - | 69.4 | 60.3 | 59.7 | 0.4 | 56.5 | 57.1 | 58.5 |
| MeetingBank | - | - | 49.0 | 40.4 | 38.4 | 38.8 | 34.8 | 39.0 |
| Py150 | - | - | - | 56.7 | 51.2 | 54.0 | 53.6 | 53.8 |
| ScienceQA | - | - | - | - | 89.5 | 64.2 | 29.5 | 54.5 |
| NumGLUE-cm | - | - | - | - | - | 33.3 | 32.1 | 33.3 |
| NumGLUE-ds | - | - | - | - | - | - | 59.7 | 57.2 |
| 20Minuten | - | - | - | - | - | - | - | 40.8 |
| **OP** | **50.0** | **59.2** | **52.6** | **51.1** | **38.5** | **48.9** | **44.6** | **48.2** |
| **BWT** | - | - | - | - | - | - | - | **-7.9%** |

Table 23: Detailed results on TRACE with GEM and HFT (start from LLAMA 2-CHAT-7B).

| Task\Round | 1 | 2 | 3 | 4 | 5 | 6 | 7 | 8 |
|---|---|---|---|---|---|---|---|---|
| C-STANCE | 50.3 | 49.0 | 47.0 | 48.3 | 50.0 | 50.7 | 50.1 | 51.3 |
| FOMC | - | 70.0 | 58.9 | 60.1 | 36.1 | 63.9 | 65.9 | 65.5 |
| MeetingBank | - | - | 47.5 | 40.2 | 38.2 | 39.2 | 39.0 | 37.9 |
| Py150 | - | - | - | 57.0 | 53.0 | 55.3 | 55.1 | 54.6 |
| ScienceQA | - | - | - | - | 88.4 | 76.8 | 70.1 | 68.4 |
| NumGLUE-cm | - | - | - | - | - | 34.6 | 24.7 | 29.6 |
| NumGLUE-ds | - | - | - | - | - | - | 60.0 | 53.6 |
| 20Minuten | - | - | - | - | - | - | - | 41.0 |
| **OP** | **50.3** | **59.5** | **51.1** | **51.4** | **53.1** | **53.4** | **52.1** | **50.2** |
| **BWT** | - | - | - | - | - | - | - | **-5.9%** |

Table 24: Detailed results on TRACE with Replay (start from LLAMA 2-CHAT-7B).

| Task\Round | 1 | 2 | 3 | 4 | 5 | 6 | 7 | 8 |
|---|---|---|---|---|---|---|---|---|
| C-STANCE | 51.7 | 50.1 | 49.4 | 48.2 | 50.6 | 49.7 | 49.9 | 52.0 |
| FOMC | - | 64.9 | 68.1 | 70.2 | 70.0 | 70.0 | 70.6 | 70.0 |
| MeetingBank | - | - | 43.4 | 48.0 | 46.1 | 46.5 | 46.4 | 44.8 |
| Py150 | - | - | - | 53.9 | 55.0 | 54.1 | 54.0 | 53.5 |
| ScienceQA | - | - | - | - | 81.9 | 86.0 | 86.3 | 87.5 |
| NumGLUE-cm | - | - | - | - | - | 30.9 | 32.1 | 32.1 |
| NumGLUE-ds | - | - | - | - | - | - | 55.7 | 53.5 |
| 20Minuten | - | - | - | - | - | - | - | 40.6 |
| **OP** | **51.7** | **57.5** | **53.6** | **55.1** | **60.7** | **56.2** | **56.4** | **54.3** |
| **BWT** | - | - | - | - | - | - | - | **1.4%** |

Table 25: Detailed results on TRACE with Replay and HFT (start from LLAMA 2-CHAT-7B).

| Task\Round | 1 | 2 | 3 | 4 | 5 | 6 | 7 | 8 |
|---|---|---|---|---|---|---|---|---|
| C-STANCE | 47.7 | 53.5 | 50.6 | 51.0 | 50.8 | 50.2 | 51.1 | 52.1 |
| FOMC | - | 61.1 | 69.4 | 70.8 | 69.8 | 70.2 | 69.4 | 69.8 |
| MeetingBank | - | - | 39.3 | 47.1 | 47.0 | 46.0 | 46.7 | 47.3 |
| Py150 | - | - | - | 55.3 | 56.3 | 56.3 | 56.5 | 55.6 |
| ScienceQA | - | - | - | - | 87.3 | 52.2 | 85.0 | 84.8 |
| NumGLUE-cm | - | - | - | - | - | 37.0 | 29.6 | 32.1 |
| NumGLUE-ds | - | - | - | - | - | - | 48.0 | 50.5 |
| 20Minuten | - | - | - | - | - | - | - | 40.5 |
| **OP** | **47.7** | **57.3** | **53.1** | **56.1** | **62.2** | **52.0** | **55.2** | **54.1** |
| **BWT** | - | - | - | - | - | - | - | **+2.1%** |

Table 26: Detailed results on TRACE with LoRASeqFT (start from LLAMA 2-CHAT-7B).

| Task\Round | 1 | 2 | 3 | 4 | 5 | 6 | 7 | 8 |
|---|---|---|---|---|---|---|---|---|
| C-STANCE | 51.6 | 48.1 | 47.4 | 46.9 | 24.1 | 12.0 | 4.1 | 7.9 |
| FOMC | - | 68.8 | 58.3 | 52.6 | 0.0 | 48.4 | 44.2 | 1.4 |
| MeetingBank | - | - | 45.7 | 10.6 | 5.9 | 1.1 | 2.7 | 3.0 |
| Py150 | - | - | - | 58.6 | 20.8 | 46.8 | 45.2 | 0.4 |
| ScienceQA | - | - | - | - | 66.1 | 50.7 | 41.3 | 0.0 |
| NumGLUE-cm | - | - | - | - | - | 33.3 | 27.2 | 0.0 |
| NumGLUE-ds | - | - | - | - | - | - | 50.5 | 0.0 |
| 20Minuten | - | - | - | - | - | - | - | 38.1 |
| **OP** | **51.6** | **58.5** | **50.5** | **42.2** | **23.4** | **32.1** | **30.7** | **6.4** |
| **BWT** | - | - | - | - | - | - | - | **-45.2%** |

Table 27: Detailed results of on TRACE with SeqFT (start from LLAMA 2-CHAT-13B).

| Task\Round | 1 | 2 | 3 | 4 | 5 | 6 | 7 | 8 |
|---|---|---|---|---|---|---|---|---|
| C-STANCE | 51.3 | 34.9 | 37.6 | 40.0 | 41.0 | 44.2 | 43.8 | 44.9 |
| FOMC | - | 70.0 | 57.5 | 52.6 | 4.2 | 49.0 | 47.2 | 49.8 |
| MeetingBank | - | - | 50.5 | 44.9 | 44.4 | 45.7 | 44.7 | 41.9 |
| Py150 | - | - | - | 56.8 | 54.9 | 54.4 | 53.1 | 54.6 |
| ScienceQA | - | - | - | - | 91.3 | 73.5 | 66.1 | 73.9 |
| NumGLUE-cm | - | - | - | - | - | 43.2 | 28.4 | 25.9 |
| NumGLUE-ds | - | - | - | - | - | - | 62.5 | 59.4 |
| 20Minuten | - | - | - | - | - | - | - | 41.4 |
| **OP** | **51.3** | **52.5** | **48.5** | **48.6** | **47.2** | **51.7** | **49.4** | **49.0** |
| **BWT** | - | - | - | - | - | - | - | **-9.4%** |

Table 28: Detailed results on TRACE with SeqFT and HFT (start from LLAMA 2-CHAT-13B).

| Task\Round | 1 | 2 | 3 | 4 | 5 | 6 | 7 | 8 |
|---|---|---|---|---|---|---|---|---|
| C-STANCE | 54.2 | 52.2 | 54.7 | 55.2 | 55.3 | 54.3 | 54.6 | 55.5 |
| FOMC | - | 73.4 | 56.7 | 54.6 | 38.3 | 43.1 | 41.9 | 50.2 |
| MeetingBank | - | - | 48.9 | 44.4 | 44.1 | 45.5 | 45.9 | 43.6 |
| Py150 | - | - | - | 58.9 | 56.3 | 56.4 | 56.7 | 56.3 |
| ScienceQA | - | - | - | - | 89.7 | 84.3 | 74.5 | 74.6 |
| NumGLUE-cm | - | - | - | - | - | 54.3 | 33.3 | 35.8 |
| NumGLUE-ds | - | - | - | - | - | - | 64.0 | 59.4 |
| 20Minuten | - | - | - | - | - | - | - | 40.9 |
| **OP** | **54.2** | **62.8** | **53.4** | **53.3** | **56.7** | **56.3** | **53.0** | **52.0** |
| **BWT** | - | - | - | - | - | - | - | **-8.5%** |

Table 29: Detailed results on TRACE with GEM (start from LLAMA 2-CHAT-13B).

| Task\Round | 1 | 2 | 3 | 4 | 5 | 6 | 7 | 8 |
|---|---|---|---|---|---|---|---|---|
| C-STANCE | 51.5 | 47.2 | 46.7 | 48.1 | 19.0 | 47.4 | 48.3 | 49.2 |
| FOMC | - | 70.5 | 59.4 | 60.2 | 0.0 | 60.7 | 58.2 | 61.2 |
| MeetingBank | - | - | 52.3 | 47.6 | 40.5 | 40.6 | 43.2 | 41.5 |
| Py150 | - | - | - | 60.7 | 60.2 | 53.6 | 54.6 | 55.7 |
| ScienceQA | - | - | - | - | 92.7 | 78.5 | 30.6 | 60.5 |
| NumGLUE-cm | - | - | - | - | - | 43.7 | 33.3 | 33.3 |
| NumGLUE-ds | - | - | - | - | - | - | 61.7 | 60.2 |
| 20Minuten | - | - | - | - | - | - | - | 41.8 |
| **OP** | **51.5** | **58.9** | **52.8** | **54.2** | **42.5** | **54.1** | **47.1** | **50.4** |
| **BWT** | - | - | - | - | - | - | - | **-8.9%** |

Table 30: Detailed results on TRACE with GEM and HFT (start from LLAMA 2-CHAT-13B).

| Task\Round | 1 | 2 | 3 | 4 | 5 | 6 | 7 | 8 |
|---|---|---|---|---|---|---|---|---|
| C-STANCE | 52.4 | 51.5 | 48.9 | 49.6 | 51.5 | 51.0 | 50.2 | 51.5 |
| FOMC | - | 73.4 | 60.8 | 61.9 | 44.4 | 65.3 | 68.9 | 67.2 |
| MeetingBank | - | - | 50.2 | 47.6 | 41.2 | 43.3 | 40.9 | 41.8 |
| Py150 | - | - | - | 61.7 | 60.1 | 60.3 | 58.7 | 57.5 |
| ScienceQA | - | - | - | - | 93.0 | 88.7 | 78.9 | 77.7 |
| NumGLUE-cm | - | - | - | - | - | 44.4 | 33.3 | 36.7 |
| NumGLUE-ds | - | - | - | - | - | - | 61.9 | 55.7 |
| 20Minuten | - | - | - | - | - | - | - | 40.6 |
| **OP** | **52.4** | **62.5** | **53.3** | **55.2** | **58.0** | **58.8** | **56.1** | **53.6** |
| **BWT** | - | - | - | - | - | - | - | **-6.1%** |

Table 31: Detailed results on TRACE with Replay (start from LLAMA 2-CHAT-13B).

| Task\Round | 1 | 2 | 3 | 4 | 5 | 6 | 7 | 8 |
|---|---|---|---|---|---|---|---|---|
| C-STANCE | 48.8 | 51.3 | 48.5 | 49.3 | 49.2 | 47.5 | 46.7 | 51.4 |
| FOMC | - | 62.3 | 70.6 | 72.4 | 71.2 | 71.2 | 70.8 | 73.0 |
| MeetingBank | - | - | 44.9 | 48.2 | 47.4 | 48.5 | 47.1 | 47.5 |
| Py150 | - | - | - | 53.9 | 55.1 | 54.2 | 47.5 | 53.3 |
| ScienceQA | - | - | - | - | 89.5 | 91.6 | 90.7 | 89.6 |
| NumGLUE-cm | - | - | - | - | - | 45.7 | 29.6 | 30.9 |
| NumGLUE-ds | - | - | - | - | - | - | 57.5 | 52.3 |
| 20Minuten | - | - | - | - | - | - | - | 39.7 |
| **OP** | **48.8** | **56.8** | **54.7** | **56.0** | **62.5** | **59.8** | **55.7** | **54.7** |
| **BWT** | - | - | - | - | - | - | - | **-0.6%** |

Table 32: Detailed results on TRACE with Replay and HFT (start from LLAMA 2-CHAT-13B).

| Task\Round | 1 | 2 | 3 | 4 | 5 | 6 | 7 | 8 |
|---|---|---|---|---|---|---|---|---|
| C-STANCE | 50.2 | 52.5 | 53.8 | 53.0 | 53.4 | 52.7 | 52.4 | 52.1 |
| FOMC | - | 61.3 | 74.2 | 71.2 | 71.8 | 73.2 | 72.4 | 73.6 |
| MeetingBank | - | - | 48.5 | 48.7 | 47.0 | 46.9 | 48.6 | 47.6 |
| Py150 | - | - | - | 55.7 | 58.2 | 55.4 | 54.0 | 54.5 |
| ScienceQA | - | - | - | - | 83.3 | 90.0 | 90.1 | 89.7 |
| NumGLUE-cm | - | - | - | - | - | 45.7 | 48.1 | 43.2 |
| NumGLUE-ds | - | - | - | - | - | - | 60.9 | 57.5 |
| 20Minuten | - | - | - | - | - | - | - | 41.0 |
| **OP** | **50.2** | **56.9** | **58.8** | **57.2** | **62.7** | **60.7** | **60.9** | **57.4** |
| **BWT** | - | - | - | - | - | - | - | **+1.6%** |

Table 33: Detailed results on TRACE with LoRASeqFT (start from LLAMA 2-CHAT-13B).

| Task\Round | 1 | 2 | 3 | 4 | 5 | 6 | 7 | 8 |
|---|---|---|---|---|---|---|---|---|
| C-STANCE | 52.4 | 44.4 | 45.1 | 39.0 | 0.0 | 41.8 | 41.1 | 12.4 |
| FOMC | - | 67.1 | 58.3 | 43.8 | 2.2 | 60.3 | 57.8 | 0.0 |
| MeetingBank | - | - | 47.3 | 11.3 | 18.2 | 14.6 | 3.2 | 12.2 |
| Py150 | - | - | - | 59.2 | 40.0 | 47.7 | 50.0 | 23.6 |
| ScienceQA | - | - | - | - | 75.4 | 70.3 | 71.0 | 67.7 |
| NumGLUE-cm | - | - | - | - | - | 47.5 | 28.5 | 25.7 |
| NumGLUE-ds | - | - | - | - | - | - | 61.3 | 28.6 |
| 20Minuten | - | - | - | - | - | - | - | 41.6 |
| **OP** | **52.4** | **55.8** | **50.2** | **38.3** | **27.2** | **47.0** | **44.7** | **26.5** |
| **BWT** | - | - | - | - | - | - | - | **-30.0%** |

