# OpenReview forum: "HFT: Half Fine-Tuning for Large Language Models"
_ICLR.cc/2025/Conference — ICLR 2025 Conference Withdrawn Submission_

### Official Review · Reviewer_68w9 · 2024-10-27

**Soundness:** 2
**Presentation:** 2
**Contribution:** 1
**Rating:** 3
**Confidence:** 4

**Summary:**

This paper proposes half finetuning as a method to solve catastrophic forgetting problem for large language models. HFT freezes half of the parameters and updating the other half. The experimental results demonstrate the effectiveness of HFT.

**Strengths:**

(1) The proposed method is simple and effective.

(2) The forgetting problem is an important problem and the authors offers a solution that tackles this problem.

**Weaknesses:**

(1) The proposed method is too simple, too heuristic without novelties. Although the authors attempt to provide some theoretical explanations on page four, they are not convincing.

(2) Why HFT outperforms FFT? Because the number of trainable parameters of HFT is less than FFT, I think FFT could be deemed as an upper bound. In my understanding, the experiments are a trade-off of balancing downstream tasks and pre-training tasks. However, for many SFT-ed checkpoints, people would care more about the downstream abilities.

(3) This paper proposes a method to mitigate forgetting. In Section 4.2, the authors compare the method with some forgetting mitigation methods but there is one important and simple baseline missing: the model average, which is a strong baseline and should be included. Could you explain why model averaging was not included in the comparisons? It would be beneficial if you could incorporate this baseline into their experimental setup to provide a more comprehensive evaluation.

**Questions:**

See weaknesses.

---

### Official Review · Reviewer_mxGw · 2024-10-28

**Soundness:** 2
**Presentation:** 3
**Contribution:** 2
**Rating:** 3
**Confidence:** 4

**Summary:**

This paper explores the problem of catastrophic forgetting in large language models (LLMs) that undergo sequential fine-tuning to enhance their ability to follow instructions and align with human preferences. Traditional full fine-tuning (FFT) can lead to the erosion of previously learned knowledge when new information is introduced. To address this, the authors propose Half Fine-Tuning (HFT), where only half of the model’s parameters are updated to learn new tasks, while the other half remain frozen, preserving earlier knowledge.

**Strengths:**

The method is simple. The paper is easy to follow.

**Weaknesses:**

1. **Lack of Distinct Advantage in PEFT Methodology**
   - The paper proposes HFT as a new PEFT method but does not convincingly demonstrate its superiority over established alternatives like LoRA, Adapters, or BitFit. While the authors claim that "HFT allows LLMs to acquire new abilities while retaining and utilizing previously learned knowledge," this characteristic is inherent to all PEFT methods, not unique to HFT.
2. **Positioning of the "Half-Reset" Technique**
   - The paper introduces the "Half-Reset" approach, yet its relevance and novelty are unclear. The method is shown to be less effective than HFT and is not new; parameter merging is already well-studied in the literature, with "Half-Reset" representing a specific case of selective parameter merging (merging at a 0.5 ratio) [1]. This technique does not constitute a meaningful contribution of the paper.
3. **Experimental Setup and Results**
   - **Unclear Benchmark Setting**: The details of the TRACE benchmark setup lack clarity. It is unclear which tasks were used to evaluate continual learning baselines, and whether task IDs are accessible for training, given that methods like LoraSeqFT exhibit significant forgetting.
   - **Lack of Baselines for Comparative Insight**: To better evaluate the performance of continual learning methods, the paper should include non-continual learning baselines, such as pooling all data into a single learning process, training individual models per task, and then ensembling their performance.
   - **Insufficient Comparative Value in Forgetting Reduction**: The results only indicate that HFT reduces forgetting compared to FFT, which is not a surprising outcome, as HFT involves modifying fewer parameters during training. Other PEFT methods would likely yield similar results.
   - **Parameter Selection Strategy in Table 4**: The choice of selection strategies in Table 4 lacks intuitive appeal. Selecting 50% of parameters at the parameter level, rather than matrix-level, layer-level, or category-level, would be more straightforward and would precisely control the selection ratio.
4. **Questionable Ease of Implementation Argument**
   - The paper claims that "HFT does not change the model architecture, making it easy to implement and scale," but this does not set HFT apart from other PEFT methods like LoRA, which are already available in easy-to-use APIs and require minimal code for implementation and scaling.

[1] Mitigating Training Imbalance in LLM Fine-Tuning via Selective Parameter Merging.

**Questions:**

Please refer to my weakness part.

---

### Official Review · Reviewer_u3vy · 2024-11-01

**Soundness:** 3
**Presentation:** 3
**Contribution:** 2
**Rating:** 6
**Confidence:** 3

**Summary:**

This paper finds that LLMs can restore some of the original knowledge by regularly resetting partial parameters. Inspired by this, the authors introduce Half Fine-Tuning (HFT) for LLMs, as a substitute for full fine-tuning (FFT), to mitigate the forgetting issues, where half of the parameters are selected to learn new tasks and the other half are frozen to retain previous knowledge. Extensive experiments and analysis on supervised fine-tuning, direct preference optimization, and continual learning consistently demonstrate the effectiveness, robustness, and efficiency of HFT.

**Strengths:**

1. The paper is well-written and easy to follow

2. The proposed method is simple and effective

3. The experiments cover different settings

**Weaknesses:**

1. Additional LLMs should be investigated to demonstrate the generalization ability of the proposed method.

2. In the continual learning setting, how does the proposed method compare to the baseline O-LoRA (Orthogonal Subspace Learning for Language Model Continual Learning)?

**Questions:**

See weakness

---

### Official Review · Reviewer_JCNY · 2024-11-04

**Soundness:** 2
**Presentation:** 2
**Contribution:** 2
**Rating:** 5
**Confidence:** 5

**Summary:**

The authors consider a special form of fine-tuning LLMs, multi-step fine-tuning, where one task is fine-tuned at a time, and after multiple rounds of fine-tuning, the entire dataset is learned. This fine-tuning mode naturally suffers from the catastrophic forgetting problem. To address the catastrophic forgetting issue, the authors propose a method where half of the parameters are frozen during one fine-tuning session while the other half are tuned. The experimental results show that the method yields slight improvements in several scenarios, including supervised fine-tuning, human preference alignment, and normal continual learning.

**Strengths:**

The method is simple, easy to implement, and can be generalized to any LLM. The experiments are sufficient and basically cover SFT and RLHF in addition to pre-training. Compared with full-parameter fine-tuning, the method proposed in this paper achieves better or comparable results.

**Weaknesses:**

The method of freezing a portion of parameters during fine-tuning is relatively common, and numerous studies have already validated the effectiveness of this freezing method. As a result, the proposed method appears a little incremental and lacks sufficient innovation. Furthermore, the effectiveness of the proposed method in the paper largely stems from the fact that the dataset size of each sub-task is relatively small. Fine-tuning a portion of parameters makes the model less prone to overfitting, rather than truly addressing the issue of catastrophic forgetting. Additionally, for LLMs, the continual learning setting, which trains only one sub-task at a time, seems rather toy-like. Such a specialized scenario is difficult to apply in practical real-world situations.

**Questions:**

Please answer the questions mentioned in the weaknesses part.

---

### Official Review · Reviewer_c6qU · 2024-11-04

**Soundness:** 2
**Presentation:** 3
**Contribution:** 2
**Rating:** 3
**Confidence:** 4

**Summary:**

This study introduces a parameter-efficient training method for fine-tuning large language models (LLMs) by limiting the number of model parameters that are updated (parameter isolation). The authors empirically choose to fine-tune 50% of the parameters and demonstrate that this approach can enhance the performance of Supervised Fine-Tuning, Direct Preference Optimization, and Continued Learning. They show that the model achieves better performance with 50% of the parameters updated compared to using all parameters.

**Strengths:**

1. This work is well-organized and easy to follow.
2. The experimental results are comprehensive and demonstrate the effectiveness of updating a subset of parameters.

**Weaknesses:**

Disadvantages:
From the view of a new fine-tuning methods, the design of the method is trivial. Numerous existing studies [1-4] have explored partial optimization (parameter isolation) to improve model performance. The heuristic selection of 50% of freezing parameters does not provide a robust framework for further advancements in the parameter selection domain. The so-called "half fine-tuning" method appears to be a specific instance of freezing selection or parameter isolation. Selection criteria could be the closeness to convergence (measure the gradient norm change between two timestamp)[1], the long-term weight history[2], block-wise periodical update history[3] and importance based soft layer sampling strategies[4].

The authors provide only limited analysis of partial parameter fine-tuning. The paper lacks a detailed examination of the importance of each layer and the choice of optimizer. Additionally, there is an absence of optimization design for parameter selection and strategies based on the selected parameters. Authors could try weight norm[4], the gradient norm change between two training steps[1] and the layer-wise parameter difference to make measurements.

In continual learning, parameter isolation [5-8] have been long-standing techniques. It would be beneficial to understand the specific differences when these are applied to LLMs. Authors could tell their methods uniqueness from previous ones by clarifying its special design compared with them and list the difference exactly for each previous methods like DERPP[5], progressive networks[6]. According to the current version, I would regard HFT as a special case of the mentioned ones.

[1] AutoFreeze: Automatically Freezing Model Blocks to Accelerate Fine-tuning, arXiv 2021

[2] SmartFRZ: An Efficient Training Framework using Attention-Based Layer Freezing, ICLR 2023

[3] BAdam: A Memory Efficient Full Parameter Optimization Method for Large Language Models, arXiv 2024

[4] LISA: Layerwise Importance Sampling for Memory-Efficient Large Language Model Fine-Tuning, NeurIPS

[5] Dark experience for general continual learning: a strong, simple baseline, NeurIPS 2020

[6] Progressive neural networks, arXiv 2016

[7] Pathnet: Evolution channels gradient descent in super neural networks, arXiv 2017

[8] Packnet: Adding multiple tasks to a single network by iterative pruning, CVPR 2018

**Questions:**

1. Why did the authors choose to update 50% of the parameters? Given the numerous possible combinations of 50% parameter selections, how do they determine which parameters to select?
2. What are the differences in the importance of each parameter when training with Direct Preference Optimization (DPO) compared to Supervised Fine-Tuning (SFT)? It would be insightful to explore methods for evaluating parameter importance rather than relying on a heuristic 50% selection.
3. Considering that DPO is typically used for alignment following SFT, what are the implications of applying parameter selection first on SFT and then on DPO?
4. How does the parameter selection strategy perform across different depths of network layers?

---

### Note · Authors · 2024-11-13

I have read and agree with the venue's withdrawal policy on behalf of myself and my co-authors.